# Tipping points in ocean and atmosphere circulations

Sina Loriani[1,2], Yevgeny Aksenov[3], David Armstrong McKay[4,5,6], Govindasamy Bala[7], Andreas Born[8], Cristiano Mazur Chiessi[9], Henk A. Dijkstra[10], Jonathan F. Donges[1,2,5], Sybren Drijfhout[11,12,13], Matthew H. England[14,15], Alexey V. Fedorov[16,17], Laura Jackson[18], Kai Kornhuber[19,20,], Gabriele Messori[21,22,23], Francesco S.R. Pausata[24], Stefanie Rynders[3], Jean-Baptiste Sallée[25], Bablu Sinha[3], Steven C. Sherwood[26], Didier Swingedouw[27], Thejna Tharammal[28]

1.  Earth Resilience Science Unit and Earth System Analysis, Potsdam Institute for Climate Impact Research, Member of the Leibniz Association, Telegrafenberg 31A, 14473 Potsdam
2.  Integrative Earth System Science, Max Planck Institute of Geoanthropology, Jena, Germany
3.  Marine Systems Modelling, National Oceanography Centre, European Way, Southampton, SO14 3ZH, United Kingdom
4.  Global Systems Institute, University of Exeter, Exeter, UK
5.  Stockholm Resilience Centre, Stockholm University, Stockholm, Sweden
6.  Geography, School of Global Studies, University of Sussex, Brighton, UK
7.  Center for Atmospheric and Oceanic Sciences, Indian Institute of Science, Bengaluru-560012, Karnataka, India
8.  Department of Earth Science, University of Bergen and Bjerknes Centre for Climate Research
9.  School of Arts, Sciences and Humanities, University of São Paulo, Av. Arlindo Bettio 1000, 03828-000 São Paulo SP, Brazil
10. Institute for Marine and Atmospheric research Utrecht, Department of Physics, Utrecht University, Netherlands
11. Royal Netherlands Meteorological Institute (KNMI), De Bilt, Netherlands
12. Ocean and Earth Science, University of Southampton, Southampton, UK
13. Faculty of Science, Utrecht University, Utrecht, Netherlands
14. Centre for Marine Science and Innovation (CMSI)
15. ARC Australian Centre for Excellence in Antarctic Science, University of New South Wales, Sydney, NSW, Australia
16. Dept of Earth and Planetary Sciences, Yale University, New Haven, CT, USA
17. LOCEAN-IPSL, Sorbonne University, Paris, France
18. Hadley Centre, Met Office, Fitzroy Road, Exeter. UK
19. International Institute for Applied Systems Analysis, Laxenburg, Austria
20. Lamont-Doherty Earth Observatory, Columbia University, New York, USA
21. Dept. of Earth Sciences, Uppsala University, Uppsala, Sweden
22. Swedish Centre for Impacts of Climate Extremes (climes), Uppsala University, Uppsala, Sweden
23. Dept. of Meteorology, Stockholm University, Stockholm, Sweden
24. Department of Earth and Atmospheric Sciences, University of Quebec in Montreal, Montreal, Quebec, Canada
25. Sorbonne Université, Laboratoire d'Océanographie et du Climat, CNRS/IRD/MNHN, Paris, France
26. Climate Change Research Centre, UNSW Sydney, Kensington NSW 2052 Australia
27. Environnements et Paléoenvironnements Océaniques et Continentaux (EPOC) Univ. Bordeaux, CNRS, Bordeaux INP, EPOC, UMR 5805, F-33600 Pessac, France
28. Interdisciplinary Centre for Water Research, Indian Institute of Science, Bangalore 560012, India

*Correspondence*: Sina Loriani (sina.loriani@pik-potsdam.de)

**Abstract.** Continued anthropogenic pressures on the Earth system hold the potential to disrupt established circulation patterns in the ocean and atmosphere. In this narrative review, we investigate tipping points in these systems by assessing scientific evidence for feedbacks that may drive self-sustained change beyond critical forcing thresholds, drawing on insights from expert elicitation. The literature provides multiple strands of evidence for oceanic tipping points in the Atlantic Meridional Overturning Circulation (AMOC), the North Atlantic Subpolar Gyre (SPG), and the Antarctic Overturning Circulation, which may collapse under warmer and 'fresher' (i.e. less salty) conditions. A slowdown or collapse of these oceanic circulations would have far-reaching consequences for the rest of the climate system and could lead to strong impacts on human societies and the biosphere.

Among the atmospheric circulation systems considered, a few lines of evidence suggest the West African monsoon as a tipping system. Its abrupt changes in the past have led to vastly different vegetation states of the Sahara (e.g. "green Sahara" states). Despite multiple potential sources of destabilisation, evidence about tipping of the monsoon systems over South America and Asia is limited. Although theoretically possible, there is currently little indication for tipping points in tropical clouds or mid-latitude atmospheric circulations. Similarly, tipping towards a more extreme or persistent state of the El Niño-Southern Oscillation (ENSO) is currently not fully supported by models and observations.

While the tipping thresholds for many of these systems are uncertain, tipping could have severe socio-environmental consequences. Stabilising Earth's climate (along with minimising other environmental pressures, like aerosol pollution and ecosystem degradation) is critical for reducing the likelihood of reaching tipping points in the ocean-atmosphere system.

# 1 Introduction

## 1.1 Tipping Points in circulation systems

The Earth's oceans and atmosphere shape both the climate and weather of the planet. Human-driven climate change is causing and will continue to cause far-reaching and long-term changes in the circulation of the oceans and atmosphere. The effect of rising greenhouse gas concentrations is to trap additional heat in the Earth system, driving atmospheric and ocean warming (with the latter accounting for more than 90 per cent of the heat trapped so far, IPCC AR6 WG1 Ch9). There is increasing evidence for changes in key circulation patterns, such as a slowing of the Atlantic Meridional Overturning Circulation (AMOC) (Dima and Lohmann 2010; Caesar et al., 2018; Rahmstorf et al. 2015; Zhu et al., 2023), and trends in mid-latitude weather patterns (Faranda et al., 2023; Horton et al., 2015; Coumou et al., 2018).

A growing body of research suggests potential tipping points in some of these systems. Tipping points are defined as critical forcing thresholds (e.g. levels of global warming or deforestation), beyond which the system switches to a qualitatively different state due to small perturbations (Lenton et al., 2008; Wang et al., 2023; Matthews et al., 2021). Here we focus on the large sub-class of systems where this change is driven by positive feedbacks in a self-sustained fashion (Armstrong McKay et al. 2022; Lenton et al., 2023). Systems with these features are called *tipping systems* (or *tipping elements* in the

73 context of large-scale Earth system components), and undergo a fundamental state transition at a rate largely determined by
74 the system itself (Armstrong McKay et al., 2022; NRC; 2002). This often occurs abruptly and/or irreversibly, with severe
consequences for the Earth and human systems (IPCC AR6 WG1 Ch4). Prominent examples involve potential widespread
dieback of the Amazon forest system, collapse of the ice sheets on Greenland and Antarctica, coral bleaching and permafrost
melting synchronised over large spatial extents. Similarly, indications for tipping can be found across the circulations in the
ocean and atmosphere: Paleoclimate proxy data suggest deep water formation in the North Atlantic has abruptly shifted to a
weaker or completely 'off' state during previous glacial cycles, with major climatic consequences – a pattern supported by
some models (Böhm et al. 2015; IPCC AR6 Ch8, 2021: Ch9 2021). Some model simulations have also shown abrupt
changes in the Antarctic deep water formation (Lago and England, 2019; Li et al., 2023). It has also been suggested that the
Indian Summer Monsoon could shift to an alternative state as a result of aerosol emissions, counter to the general trend of
monsoon strengthening with warming (Levermann et al. 2009; IPCC AR6 WG1 Ch10), as well as potential shifts in
circulations on the Southern Hemisphere to El Niño-like mean conditions (Fedorov et al, 2006). Additionally, numerical
simulations point to the possibility of tipping behaviour in atmospheric blocking, in the form of a self-sustaining,
feedback-driven shift (Drijfhout et al., 2013). Furthermore, there have been discussions on potential large-scale
reorganisation of tropical circulation and cloud systems (Schneider et al., 2019, Caballero and Carlson, 2018).

**1.2 Assessment approach**

In this review article, we assess the available literature on potential tipping points across the outlined wide range of
circulation systems. This work contributes to recent efforts aimed to categorise the breadth of Earth system components with
respect to their potential for tipping across the different domains, and has served as the corresponding chapter for the
recently published Global Tipping Points Report (Lenton et al., 2023), complementary to chapters on potential tipping points
in the cryosphere and biosphere. Our assessment draws on several sources of evidence: Direct observations via in-situ
measurements or remote sensing allow for near-time monitoring of the systems, and can provide indications that the system
is weakening or even showing early warning signs of approaching a tipping point. However, these observational time series
are often (too) short to observe nonlinear behaviour in potential tipping systems. Observational and paleoclimate proxies
(i.e. indirect reconstructions) greatly extend data availability to longer time scales and offer historical analogues of tipping
dynamics. They are complemented by numerical models of varying complexity, which are used to study system-wide
feedbacks in a consistent setup. Concerning oceanic and atmospheric circulations, the different generations of
Atmosphere-Ocean General Circulation Models (AOGCMs) up to state-of-the-art coupled Earth System models have been
providing great insights into the dynamics of these systems, not least via standardised assessments in the context of the
Coupled Model Intercomparison Project (CMIP, Eyring et al., 2016). Eventually, however, these models are limited in
resolution and can only resolve potential tipping dynamics if the model captures the respective underlying feedback.
Therefore, conceptual and low-order models are indispensable to guide the understanding of potential tipping mechanisms

and, potentially, to identify modelling gaps. In summary, all the outlined lines of evidence complement one another with their respective strengths and drawbacks, and increase confidence in a system's assessment where they converge (Boers et al., 2022).

For our assessment, we therefore draw on a rating system where confidence increases with both robustness (high robustness = multiple, consistent, independent lines of high-quality evidence) and agreement of evidence, as developed and employed in the IPCC context (Mastrandrea et al., 2010). To this end, a system is **classified as tipping system** in this work with

- **high confidence (+++):** Multiple, independent lines of evidence consistently indicate the presence of feedback loops that can drive self-perpetuating change beyond a threshold, leading to a state shift in that system. E.g., there are strong paleo analogues, consistent tipping behaviour in models across the hierarchy, and, if applicable (when the tipping threshold is expected at present-day or soon-to-be-reached forcing levels), proxy and direct observations are compatible with the expected tipping dynamics.

- **medium confidence (++):** Multiple, independent lines of evidence indicate the presence of such feedback loops, however there are uncertainties in timing, magnitude or feedback strength. E.g., there are tipping dynamics in some models, and paleo records hint at dynamics compatible with tipping. Support from observations is limited or contested.

- **low confidence (+):** Singular lines of evidence indicate the presence of such feedback loops, and tipping dynamics only emerge in specific models or under constrained assumptions. E.g., tipping is in principle conceivable via conceptual models, but there are no clear or only weak paleo analogues, and limited demonstration of tipping dynamics in numerical models.

Similarly, we classify a system as **not being a tipping system** if there is evidence indicating the lack of feedback loops that can drive self-perpetuating change beyond a threshold (with low/medium/high confidence). It is **unclear** whether the system is a tipping system if there is conflicting or limited evidence about the existence of such feedback loops. Note that in this approach, irreversibility and abruptness are useful indicators for tipping dynamics, but the defining criterion is a feedback loop beyond a forcing threshold as outlined above, extending the IPCC definition of a tipping point (*"A critical threshold beyond which a system reorganizes, often abruptly and/or irreversibly"*; Matthews et al., 2021) by a mechanistic understanding (see also Armstrong McKay et al., 2022) reflecting the notion of a catastrophic bifurcation with a vanishing stable attractor (Dijkstra, 2013). Furthermore, it is important to emphasise that our assessment reflects the confidence in determining whether a system *can* tip (under present-day or broadly plausible future conditions) — not the confidence that it *will* tip. The latter depends on the trajectory of external forcing and the position of the critical thresholds, both of which are subject to considerable uncertainties.

Although tipping point research is expanding, systematic model intercomparison studies dedicated to tipping dynamics remain scarce—limited so far to isolated efforts like the North Atlantic Hosing Modelling Intercomparison Project, (NAHosMIP; Jackson et al., 2023) or the Ice Sheet Model Intercomparison Project (ISMIP; Nowicki et al., 2016) targeting specific aspects of one system. As a result, current assessments like ours can offer rather qualitative confidence statements

as outlined above, while highlighting the need for future work to establish a more comprehensive quantitative foundation. In
the future, emerging initiatives like the Tipping Points Modelling Intercomparison Project (TIPMIP; Winkelmann et al.,
2025) could enable confidence ratings based on the fraction of models that exhibit abrupt shifts in setups dedicated to
scrutinise tipping dynamics. For now, this review contextualises the current state of science through the lens of
self-reinforcing feedbacks beyond critical forcing thresholds, with a focus on consistency and qualitative agreement across
lines of evidence.

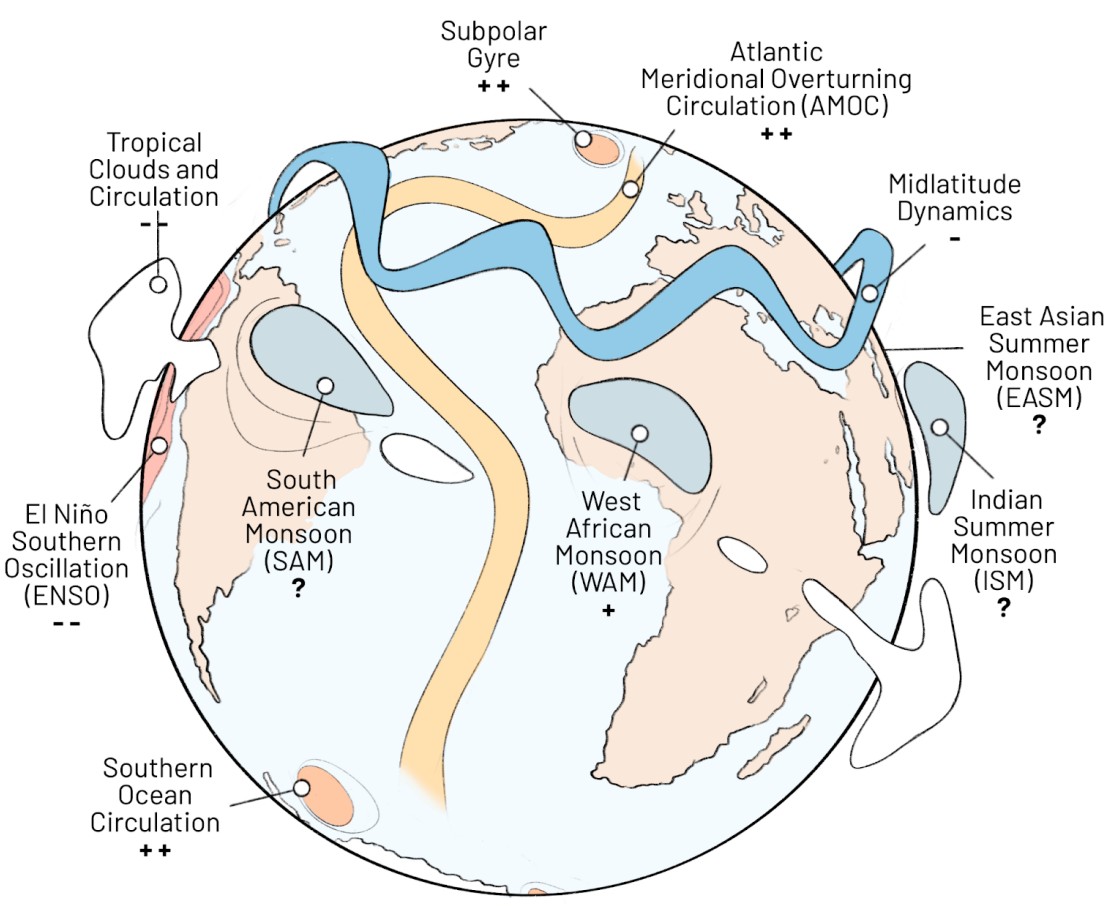

**Figure 1: Potential tipping systems in ocean and atmosphere circulations considered in this review. +/++/++ denotes that a system**
**is classified as a tipping system with low/medium/high confidence; "?" marks conflicting or limited evidence.**

## 2 Ocean Circulation

### 2.1 Atlantic Meridional Overturning Circulation (AMOC)

The Atlantic Meridional Overturning Circulation (AMOC) refers to a three-dimensional circulation present in the Atlantic (Figure 2a) whereby warm upper ocean waters ('upper branch') move northward from the tip of Southern Africa to the northern North Atlantic, where they cool, sink and return southwards as cold deep waters ('lower branch'). The AMOC shapes the climate of the whole Earth, influencing, for example, the 1-2°C temperature difference between the Northern and Southern Hemispheres, and the location and strength of rainfall across all tropical regions (Buckley and Marshall, 2016, Feulner et al., 2013, Marshall et al., 2014).

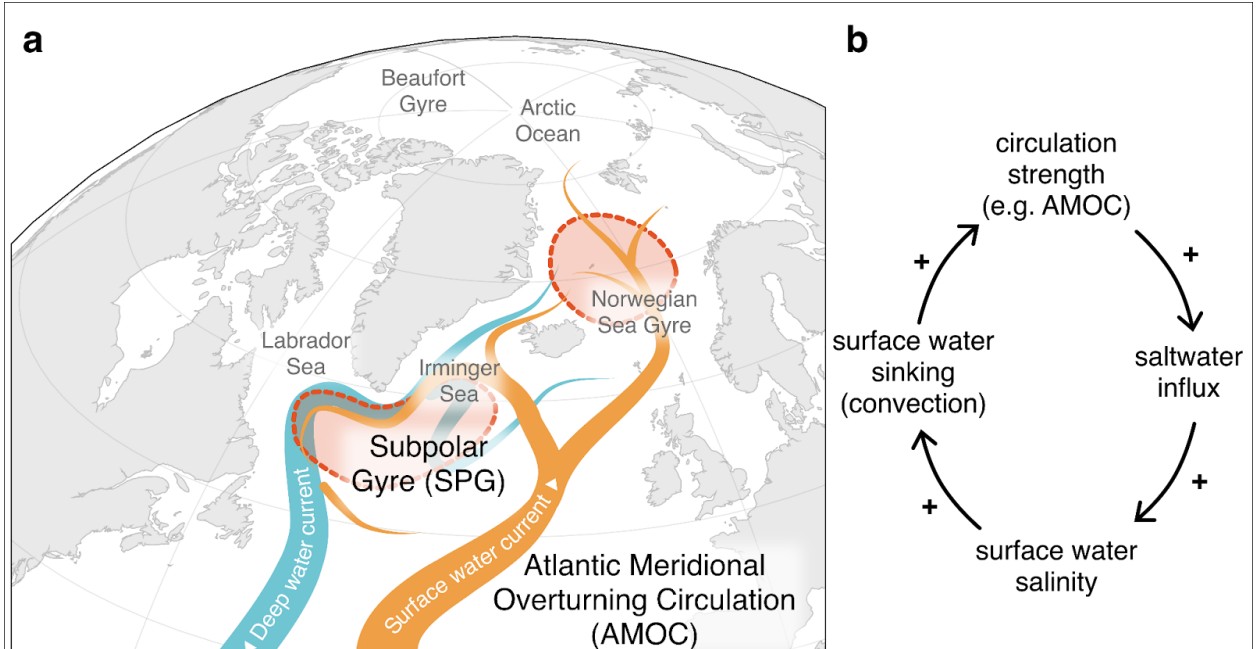

**Figure 2: Overview of the major oceanic circulation systems in the North Atlantic. a) The surface currents (orange pathways) are connected to deep ocean currents (blue) through sites where dense (cold, salty) water sinks, driving the overturning circulation (pink shaded areas). b) One critical feedback is the salt-advection feedback, in which a weakening of the circulation strength reduces the northward advection of salt, which in turn weakens the circulation.**

Fresh, warm water is less dense than cold, salty water. In the future, surface waters in the northern North Atlantic will become less dense, because of anthropogenic warming and freshening from ice melt and increased regional precipitation. This will weaken deep water formation in that region, which will disrupt the AMOC, causing it to weaken significantly or even collapse completely. AMOC strength has only been observed directly since 2004 (Srokosz and Bryden, 2013), with more uncertain reconstructions based on observations such as surface temperature, which extend back in time before 2004 ('observational proxies'), or from paleoclimate archives such as ocean sediment cores which extend back to prehistoric times ('paleoclimate proxies') (Caesar et al. 2018, 2021, Moffa-Sánchez et al., 2019). The lack of a sufficiently long

observational record is a major hurdle for robust understanding of the AMOC, particularly since the North Atlantic is a region of high variability on interannual to decadal timescales (Boer 2000). Current observations show that it is freshening at subpolar latitudes (50-65°N), most strongly in the upper 100m, and warming, most strongly between 100-500 m water depth (IPCC AR6, 2021). Both trends act to reduce AMOC strength. Greenland Ice Sheet melt is accelerating and releasing extra fresh water into the North Atlantic (The IMBIE Team, 2020). In addition, Arctic sea ice is reducing in surface extent and thickness (Serreze and Meier, 2019) and overall Arctic river discharge is increasing (Holmes et al., 2021), adding fresh water to the Arctic which can then leak into the North Atlantic.

Limited direct observations of AMOC strength make current trends uncertain, but there are some indirect signs of ongoing weakening. Observational and paleoclimate proxies suggest the AMOC may have weakened by around 15 per cent over the past 50 years (Caesar et al. 2018) and may be at its weakest in 1,000 years (Caesar et al., 2021). However, the proxy data used in these studies have large uncertainties, and agreement amongst reconstructions is contentious (Moffa-Sanchez et al., 2019, Killbourne et al., 2022, Caesar et al., 2022, Terhaar et al., 2025). It remains therefore difficult to confidently discern potential recent trends from natural variability (Bonnet et al. 2021, Latif et al., 2022) since natural variability is still able to explain the trend over the last century. What is clear is that if the observed cooling trend in the subpolar gyre is only due to external forcing, then CMIP6 climate models do not capture correctly such a negative trend in their ensemble mean (cf. Figure 2 of Qasmi, 2022).

The IPCC's most recent assessment is that the AMOC has weakened relative to 1850-1900, but with low confidence due to disagreement among reconstructions (Moffa-Sanchez et al., 2019, Killbourne et al., 2022) and models forced response (IPCC AR6 Ch9, 2021). For the future, the IPCC projects that it is very likely that the AMOC will decline in the 21st century (however with low confidence on timings and magnitude) (IPCC AR6 Ch4, 2021). There is medium confidence (about 5 on a scale of 1 to 10) that a collapse would not happen before 2100, though a collapse is judged to be as likely as not by 2300. Hence the possibility of an AMOC collapse within the next century is very much left open by the latest IPCC report.

## 2.1.1 Evidence for tipping dynamics

The AMOC has been proposed as a 'global core' climate tipping system with medium confidence by Armstrong McKay et al. (2022). Paleorecords indicate it may have abruptly switched between stronger and weaker modes during recent glacial cycles (Figure 3). Most of the time (including the warm Holocene of the past 12,000 years) the AMOC is in a strong, warm mode, but during peak glacials it sometimes shifted to a weak, cold mode instead (Böhm et al., 2014). It may also occasionally have collapsed entirely to an 'off' mode during some 'Heinrich' events, in which iceberg outbursts from the North American Laurentide Ice Sheet temporarily blocked Atlantic overturning for several centuries (Alley et al., 1999; Rahmstorf et al. 2002; Weijer et al. 2019). While more evidence exists for switches between a warm and cold on-mode in the paleo records, some of these data support switches to an off-mode, possibly occurring through a two-step process from warm-on to cold-on to off (Weijer et al., 2019).

In two previous censuses of climate model projections, a shut-down of the AMOC was observed in a small minority of simulations (Drijfhout et al., 2015; Sgubin et al., 2017). This shut-down took many decades and was preceded by decreases

in subpolar surface air and ocean temperatures and increased sea ice cover. Ultimately, deep mixing ceased to occur, cutting

the connection between the surface and the deep ocean. Although only a few climate models showed a shutdown of the

AMOC, there are concerns that the AMOC may be too stable in CMIP-type climate models (Mecking et al., 2017, Liu et al.,

2017), and hence may underestimate the likelihood of AMOC collapse (IPCC AR6 WG1 Ch9).

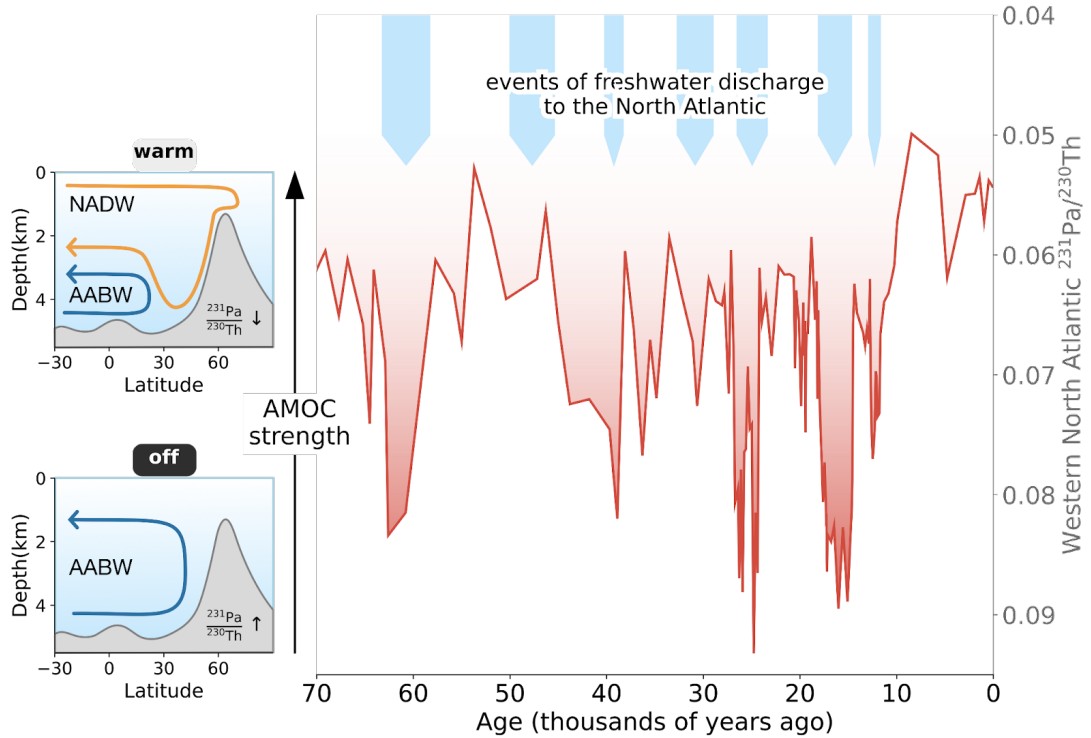

**Figure 3: Different AMOC modes and paleo-evidence. The diagrams on the left show two AMOC modes as indicated by**
**sedimentary 231Pa/230Th in paleo records. NADW: North Atlantic Deep Water; AABW: Antarctic Bottom Water, adapted from**
**Böhm et al., 2015. B AMOC slowdown events during the last 70,000 years as recorded by sedimentary 231Pa/230Th data**
**(McManus et al., 2004; Böhm et al., 2015) from the Western North Atlantic (Bermuda Rise, ca. 34oN). Sedimentary 231Pa/230Th**
**from the Bermuda Rise is a proxy for AMOC strength that assesses the southward flowing North Atlantic Deep Water between ca.**
**3,500 and 4,500m water depth. The top of the panel marks the timing of past major events of freshwater discharge to the high**
**latitudes of the North Atlantic that decreased AMOC strength (Sarnthein et al., 2003; Carlson et al., 2013; Sanchez Goñi and**
**Harrison, 2009). The red shading highlights past AMOC slowdown events. There is also evidence of AMOC shifts during the last**
**interglacial period, 116,000-128,000 years ago (Galaasen et al. 2014).**

Some recent studies have suggested that early warning signals' indicating destabilisation can be detected in reconstructed

'fingerprints' of AMOC strength over the 20th Century (Boers, 2021), and if a tipping point is assumed then the collapse

threshold could be reached during the 21st Century (Ditlevsen & Ditlevsen, 2023). These studies used observational proxies

for temperature and salinity from the Northeastern subpolar North Atlantic, which are used as indirect AMOC fingerprints

rather than direct measurements of AMOC strength. This gives long enough data to analyse for early warning signals, but

using indirect proxies adds uncertainty. For instance, the models used by Boers (2021) and Ditlevsen & Ditlevsen (2023) to

project collapse are highly simplified, using an indirect 1D metric for the AMOC with a double-well as its stability

landscape assumed, and do not take into account the low-frequency variability of the AMOC. After fitting the timeseries of
the metric to various statistical properties of a 1D variable approaching a tipping point (switching from one well to the other
in its stability landscape), there are substantial uncertainties with this methodology (see also Michel et al., 2023 highlighting
potential false warnings) in particular to determine tipping times (Ben Yami et al. 2024). Despite these caveats, the results
amount to a serious warning that the AMOC might be en route to tipping, supported by further potential early warning
signals found from analysis of Northern Hemisphere paleoproxies (Michel et al., 2022), and as derived from a physics-based
signal (van Westen et al., 2024). However, the claim that we might expect tipping within a few decades is not substantiated
enough.

AMOC stability is strongly linked to the 'salt-advection feedback' (Stommel, 1961, and see Figure 2b). The AMOC imports
salt into the Atlantic and transports it from the South Atlantic to the northern North Atlantic. If the AMOC weakens then
less salt is transported to the northern North Atlantic, the surface waters freshen, which inhibits sinking, and the AMOC
weakens further. The AMOC collapses seen in models (Drijfhout et al., 2015, Sgubin et al., 2017) were driven by this
salt-advection feedback. However, the strength of this feedback, and the timescale over which it operates are governed by
processes whose effects are quite uncertain. Although Figure 2a shows typical pathways of surface and deep water through
the Atlantic, these pathways are an average picture over many decades. Individual water parcels may get caught up in
basin-scale surface or deep recirculations, smaller-scale eddies and meandering currents. There is no definitive evidence
though from models or observations that these small-scale processes systematically impact the salt advection feedback.

Additionally, changes in the AMOC have other impacts on salinity – for instance through affecting evaporation and
precipitation patterns (Jackson 2013, Weijer et al, 2019). These other feedbacks can temporarily mask, and may even
overcome, the salt-advection feedback, potentially changing the stability of the AMOC (Jackson, 2013, Gent, 2018). It is
difficult to characterise these processes and feedbacks from observations alone due to insufficient data coverage both in time
and space, so we are dependent on numerical models. However, many studies have used reduced complexity models, which
may not capture all the potential feedbacks, and even the current generation of climate models have quite low spatial
resolution and do not characterise narrow currents, eddies and processes such as horizontal and vertical mixing very well.
Indeed, it has been shown recently (Swingedouw et al. 2022) that the inclusion of fine-scale processes of the order of a few
kilometres can increase the spread of Greenland meltwater from coastal waters towards the centre of the Labrador Sea,
which might more rapidly affect deep convection processes through reduction in surface salinity.

Armstrong McKay et al (2022) estimated with low confidence a global warming threshold for AMOC collapse of ~4°C
(1.4-8°C). In our view, the range is a better indication of the uncertainty in the different model responses rather than a
relationship to global warming, as the likelihood is probably less dependent on temperature, but strongly depends on salinity
changes and the strength of opposing feedbacks on the freshwater budget. Studies with climate models have found that
adding freshwater can cause the AMOC to collapse and not recover in some models. Since many climate models might be
biased towards stability, however, these studies use an unphysically large amount of freshwater to explore the sensitivity
(Jackson et al., 2023). Although adding freshwater causes a collapse, they show the threshold is dependent on the strength of

the AMOC and deep convection, rather than on the amount of freshwater added (Jackson and Wood 2018, Jackson et al,
2023). AMOC collapse may also be more sensitive to the rate of freshwater forcing than the total magnitude (Lohmann &
Ditlevsen, 2021).

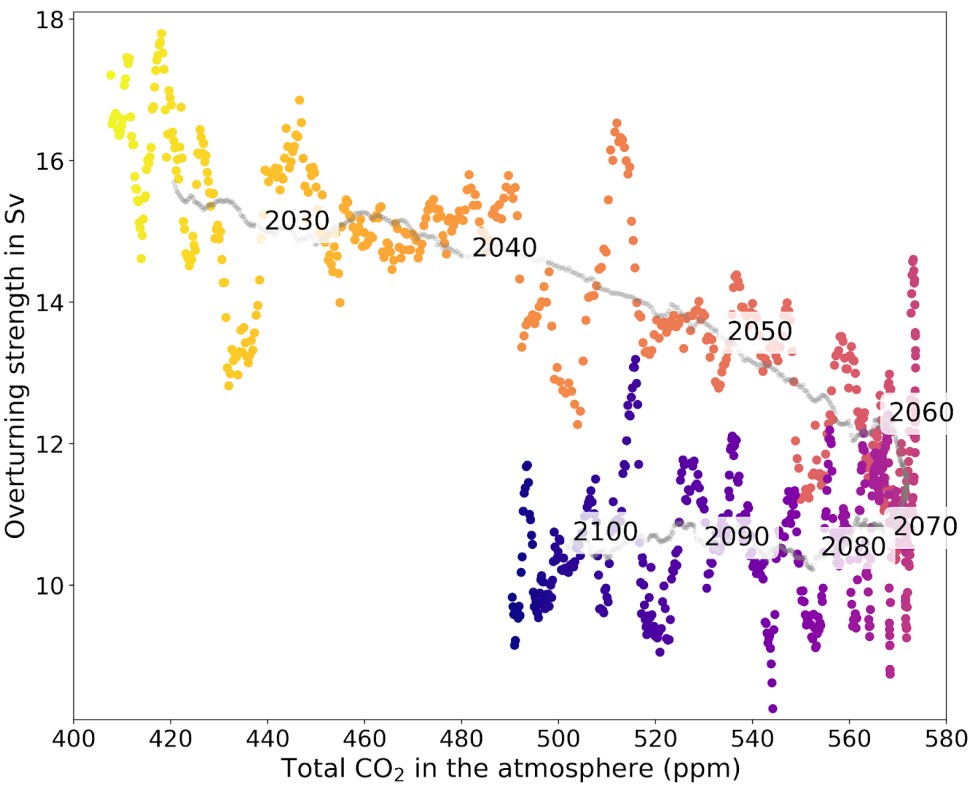

**Figure 4: AMOC hysteresis in a CMIP model, adapted from Heinze et al. (2023). CMIP6 overshoot experiments (using UKESM;**
**Jones et al. 2020) showing hysteresis - different states of the AMOC (vertical axis) for the same atmospheric CO2 concentration**
**(horizontal axis). Possible causes are delayed or non-linear response to forcing or possibly bistability of AMOC. The AMOC**
**strength is measured in 'Sverdrups' (Sv; i.e. a flow of 1 million cubic metres per second); colours from yellow to blue show model**
**years from 2015 to 2100 respectively.**

Hysteresis and bistability both refer to systems which can adopt one of two or more statistically stationary states for the
same external forcing, such as CO2 concentration (see Figure 4 and Boucher et al., 2012). Commonly, this is explored by
approaching the same external conditions with different trajectories in model simulations, e.g. increasing and reversing the
forcing to study reversibility. Bistability involving a full collapse of the AMOC by artificially flooding the North Atlantic
with freshwater has been demonstrated (or strongly implied) in theoretical models (Stommel, 1961) and climate models of
reduced complexity (Rahmstorf et al., 2005; Hawkins et al., 2011). These types of numerical experiments study bistability
through forcings that change slowly enough for the system to equilibrate, typically requiring long simulations and thus
coarse model resolution for reasonable computational performance. Alternatively shorter experiments, where forcing is
applied temporarily, have been conducted with more complex models. These have demonstrated weak AMOC states which
persist for at least 100 years in about half of a test group of CMIP6-type models (Jackson et al., 2023) and in a

high-resolution ocean-atmosphere coupled climate model (Mecking et al., 2016). In simpler (coarser resolution models) full hysteresis experiments have been carried out and those unequivocally support the existence of bi-stability (Weijer et al 2019). Only recently has AMOC tipping (Van Westen et al., 2025) and a bistable regime (Van Westen et al., 2023) been found in a CMIP-type model in response to gradually changing increasing freshwater release in the North Atlantic. Hence, AMOC tipping has been found over a whole hierarchy of ocean-climate models, still only excluding very high-resolution (strongly eddying) climate models, for which appropriate experiments have not yet been conducted.

Besides a bi-stable regime bounded by two bifurcation points, a mono-stable off- and on-mode exist on each side of the two bifurcation points. In present-day climate models through tuning the AMOC always is characterised by the warm on-mode and either resides in the monostable or bistable regime. Whether or not the AMOC is stable under pre-industrial and present-day conditions must be model-dependent, depending on details of the background state, probably to a large extent determined by the freshwater balance in the Atlantic and how northward salinity or freshwater transport by the ocean circulation balanced by the surface forcing. It is not yet understood why transitions to an off-state or much weakened AMOC state forced by the same freshwater hosing occurs in some models and not in others (Jackson et al., 2023). However, as previously mentioned, there is evidence that the present generation of climate models is too stable due to model biases in the distribution of ocean salinity (Liu et al., 2017, Mecking et al 2017). Not only is it difficult to prove system bistability, the complexity of the system and interaction with multiple drivers make it hard to assess collapse thresholds. It may be that realistic freshwater input is not sufficient to cause the transition, or that changing $CO_2$ alters the underlying system stability, thus increasing the critical freshwater threshold (Wood et al., 2019). Nevertheless, overshoot scenarios, where the $CO_2$ trend is assumed to reverse at some point in the future, provide some useful information about reversibility of the AMOC on human timescales (e.g. decades to centuries). Figure 4 shows how the AMOC changes in the UKESM climate model under an overshoot emission scenario exceeding and returning to 500ppm. Even if $CO_2$ concentrations return to 500ppm by 2100 the AMOC is still only 77 per cent of the strength it was in 2050 also at 500ppm, exhibiting irreversible weakening on these timescales.

The timescale of AMOC tipping was estimated by Armstrong McKay et al (2022) to be 15-300 years, however this range is very dependent on the strength of the freshwater forcing applied in experiments, which in many cases is unrealistically large as compared to projected melting of the Greenland Ice Sheet and increase in precipitation and river runoff. Moreover, the assessment is also potentially impacted by the models being unrealistically stable. With a realistic forcing scenario, the timescale will depend on the feedbacks. A basin-wide salt advection feedback may have a century timescale, while the same salt feedback acting on a regional scale (between North Atlantic subtropical and subpolar gyre) is only a few decades. The latter feedback then may induce fast tipping to a (probably) metastable state that on the longer term is stabilised or destabilised by the basin-wide Atlantic salt feedback. Even faster timescales associated with tipping to such a hypothetical metastable state are possible when deep mixing is capped off by sudden increases in sea ice cover, or freshwater halocline. This fast tipping is then induced by convective feedbacks after which the advective salt feedbacks may further stabilise or destabilise the convection collapse (Rahmstorf et al., 2001; Kuhlbrodt et al., 2001).

AMOC collapse would lead to cooling over most of the Northern Hemisphere, particularly strong (up to 10°C relative to preindustrial) over Western and Northern Europe. In addition, a southward shift of the intertropical convergence zone would occur, impacting monsoon systems globally and causing large changes in storminess and rainfall patterns (Jackson et al, 2015). A collapse of the AMOC would influence sea level rise along the boundaries of the North Atlantic, modify Arctic sea ice and permafrost distribution (Schwinger et al., 2022; Bulgin et al, 2023), reduce oceanic carbon uptake (Rhein et al., 2016) and potentially lead to ocean deoxygenation (Kwiatkowski et al., 2020) and severe disruption of marine ecosystems (including changes in the North Atlantic Subpolar Gyre, see below), impacting North Atlantic fish stocks (e.g. Heinze et al., 2021).

## 2.1.2 Assessment and knowledge gaps

Although the AMOC does not always behave like a tipping system in many ocean/climate models, paleoceanographic evidence strongly points to its capability for tipping or at least to shift to another state that can be quasi-stable for many centuries (Figure 3). Tipping is also suggested in a recent study of several CMIP6 models (Jackson et al, 2023) and in another study which found that removing model salinity biases strongly increased the likelihood of tipping (Liu et al., 2017). This does not necessarily mean that tipping is likely in a future climate, since some of these scenarios specified unrealistic inputs of freshwater or GHG emissions. Nonetheless, although the likelihood for collapse is considered small compared to the likelihood of AMOC decline, the potential impacts of AMOC tipping make it an important risk to consider in framing mitigation targets, for instance.

The latest AR6 assessment states that we have only medium confidence that an AMOC collapse will not happen before 2100 (IPCC AR6 WG1 Ch9). This uncertainty is due to models having strong ocean salinity biases, absence of meltwater release from the Greenland Ice Sheet in climate change scenarios, and the possible impact of eddies and other unresolved ocean processes on freshwater pathways. However, a recent study with the PAGES2K database of climate reconstructions of the past 2,000 years suggests, using statistical methods based on dynamical systems theory, that the AMOC variability might be slowing down (Michel et al., 2022), as do the studies of Boers (2021) and Ditlevsen and Ditlevsen (2023). If an AMOC tipping does exist, as suggested in a number of models (e.g. Boulton et al. 2014), this might mean that the AMOC system might be approaching this tipping point. AR6 also concluded that reported recent weakening in both historical model simulations and observation-based reconstructions of the AMOC have low confidence. Direct AMOC observations have not been made for long enough to separate a long-term weakening from short-term variability. Another recent study suggests that we will need to wait until at least 2028 to obtain a robust statistical signal of AMOC weakening (Lobelle et al. 2020). Thus, the coming years will be crucial for detection of an AMOC weakening potentially leading to longer-term instability.

There are substantial uncertainties around how the AMOC evolves over long timescales, because of a lack of direct observations. More paleo-reconstructions of AMOC strength, ocean surface temperature, and other AMOC-related properties with high temporal resolution, using appropriate proxies and careful chronological control performed for key past periods (e.g. last millennium, millennial-scale climate change events, previous interglacials), hold great potential to improve

our understanding about the AMOC as a tipping point. Indeed, this might help to test early warning metrics coming from
dynamical system theory in some past real-case conditions. These reconstructions might be based on several sources of
proxy data to gain robustness (e.g. sortable silt in the deep ocean, 231Pa/230Th in the deep ocean, radiocarbon in the deep
versus surface ocean, surface and intermediate ocean fingerprints). Other open issues are to: (i) reconcile disagreements
between paleo-reconstructions and model simulations, and (ii) develop improved fingerprints for creating historical
reconstructions and monitoring the AMOC.

Current climate models suffer from imperfect representation of some important processes (such as eddies and mixing) and
from biases which can impact the AMOC response to forcings. Hence we need to assess how important these issues are for
representing AMOC stability, in particular, to understand how different feedbacks vary across models and are affected by
modelling deficiencies. Given these issues, a robust assessment of the likelihood of an AMOC collapse is difficult, but based
on the evidence presented, we assess that the AMOC features tipping dynamics with medium confidence. One potential way
forward, given these uncertainties, is in developing observable precursors to a collapse that could be monitored.

**2.2 North Atlantic Subpolar Gyre (SPG)**

The North Atlantic Subpolar Gyre (SPG) is an oceanic cyclonic (counter-clockwise in the northern hemisphere) flow to the
south of Greenland (Figure 5), driven by wind and buoyancy forcing. The latter is linked to a site of deep ocean convection
in the Labrador-Irminger Seas, i.e. sinking of the subsurface ocean waters to great depths, contributing to the AMOC
(Figures 2, 5, 6). Some studies have suggested that the SPG could have a tipping point where the convection collapses,
having profound impacts on the SPG system.
There are indications for change in the SPG, as observations show that Labrador Sea Water (LSW) formed during oceanic
deep convection events after 2014 was less dense than the LSW formed between 1987 and 1994 (Yashayaev and Loder,
2016), potentially influencing the AMOC. Moreover, the observed 'warming hole' over the North Atlantic can be explained
by AMOC slowdown (Drijfhout et al 2012, Caesar et al 2018, also see AMOC above) and has also been linked to SPG
weakening in CMIP6 models (Sgubin et al 2017, Swingedouw et al 2021). In these models, a collapse of the oceanic
convection causes a localised North Atlantic regional surface air temperature drop of ~2-3°C. This cooling moderates
warming over north-west Europe and eastern Canada in global warming scenarios, although it is smaller and less widespread
than that associated with AMOC collapse.

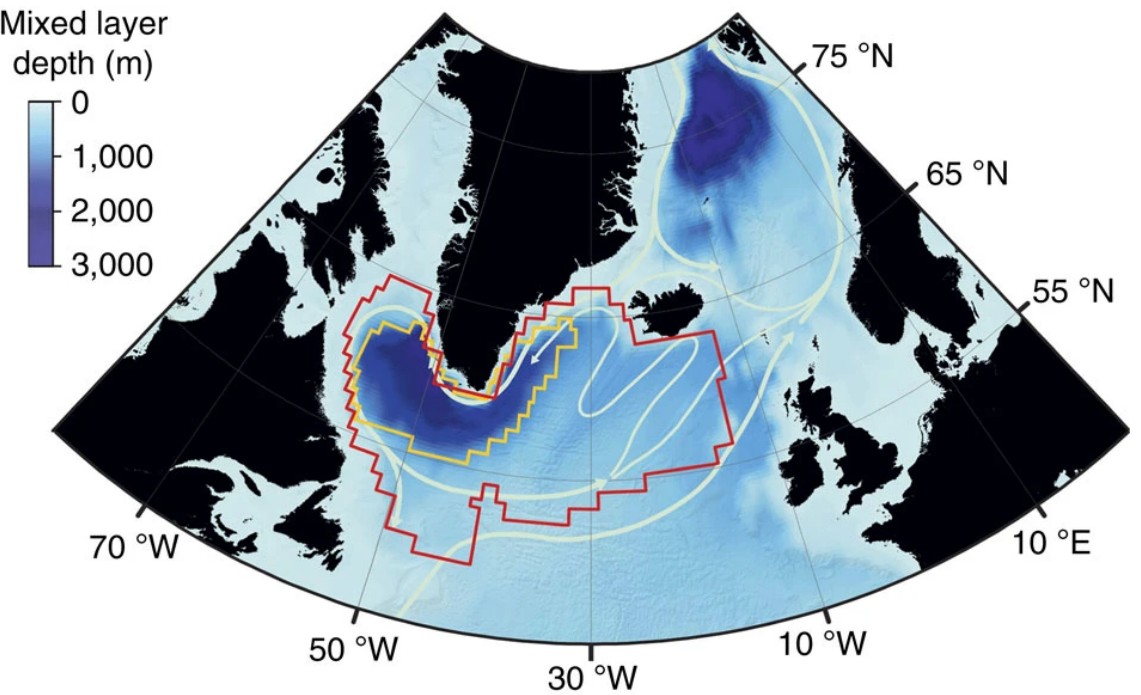

**Figure 5: Map showing the maximum ocean mixing depth in the North Atlantic (light to dark blue), showing deep water convection sites driving the AMOC and SPG east and south of Greenland respectively (with the Labrador-Irminger Seas convection area bordered by yellow). The pale arrows show surface water currents, with the anti-clockwise subpolar gyre occurring within the red bordered area. From Sgubin et al. (2017).**

A northward-shift of the atmospheric jet stream, which is predicted to take place with SPG weakening, means more weather extremes in Europe (which may be linked to the unusual cooling and heat waves in recent years) (Osman et al., 2021) and southward shift of the intertropical convergence zone (ITCZ, see Figure 1) (Sgubin et al. 2017; Swingedouw et al. 2021). Models suggest potential impact of the SPG collapse on the European weather, precipitation regime and climate (Swingedouw et al., 2021). This change in the physical system may trigger changes in ecosystems with detrimental consequences for the North Atlantic spring bloom and the overall Atlantic marine primary productivity. Neither of these reverse to the preindustrial state even when the emissions do by the 2100s as the models show (Yool et al., 2015; Heinze et al., 2021). This would impose a strong impact on fisheries and biodiversity (Swingedouw et al., 2021), with expected wide societal implications (e.g., Holm et al., 2022). Last but not least, a transition between two SPG stable states has been suggested to explain the onset of the so-called 'Little Ice Age' in which colder conditions prevailed in Europe during the 16th-19th centuries (Lehner et al. 2013; Schleussner et al., 2015; Michel et al. 2022).

Ventilation of Labrador Sea Water (LSW) is accompanied by an uptake of oxygen. Starting in 2014, the convection in the Labrador Sea became more intense and reached depths of 1,500m and below. Consequently, oxygen in LSW is in general increased, but this increase did not penetrate the densest part of this water mass (Rhein et al., 2017). The oxygen concentrations in the deepest part of the LSW (around 2,000m) have decreased in the formation region and along the main export pathways (southward and eastward crossing the Mid-Atlantic Ridge) for more than 20 years. Most of the oxygen

from the export of newly formed LSW has been consumed north of the equator (Koelling et al., 2022), and the long-term oxygen decline along the southward LSW pathway might have impacts on ecosystems in the tropics and subtropics over longer timescales (e.g., Heinze et al., 2021).

The potential shutting-down of winter convection in the Labrador Sea (see Figure 6a,b and Swingedouw et al., 2021) will also stop the production of Labrador Slope Water (LSLW). This water is next to the Labrador Sea continental slope and is lighter and less deep than LSW. It contributes to AMOC and the Gulf Stream and can influence variability of the Atlantic climate system overall (New et al., 2021). The LSLW is rich in nutrients and oxygen too, thereby affecting the ecosystems on the North American continental shelf and shelf slope (e.g. Claret et al., 2018) and might affect tropical and subtropical marine ecosystems on a timescale of several decades. Furthermore, the SPG takes up large amounts of atmospheric carbon and exports it to the deep ocean (Henson et al., 2022).

Shallowing of the SPG (Sgubin et al., 2017; Swingedouw et al., 2021) would directly increase regional $CO_2$ uptake but negatively impact marine biology, for instance threatening the habitat of cold-water corals in the area due to higher acidity with more $CO_2$ dissolved in the water (Fröb et al., 2019; Fontela et al., 2020; Garcia-Ibanez et al., 2021). Weakening or collapse of the SPG would reduce the amount of carbon-depleted intermediate water being upwelled and newly carbon-enriched water being convected, reducing export of anthropogenic $CO_2$ to the deep ocean (Halloran et al. 2015; Ridge and McKinley 2020), which in turn might lead to an increase of atmospheric $CO_2$ concentration on the long term (Schmittner et al. 2007). Declining SPG strength may also be reducing the currently high phytoplankton productivity in this area (Osman et al., 2019; Henson et al. 2022), reducing the amount of biologically fixed carbon to deeper water too.

Changes in the overall Atlantic ocean circulation (AMOC and SPG) can impact the spread of Atlantic water into the Arctic and affect marine ecosystems there. Summer sea ice decline reduces light limitation, rendering Arctic ecosystems more similar to the present North Atlantic (Yool et al., 2015). Increased seasonal phytoplankton blooms will deplete nutrients in the ocean, but increased inputs from rivers and coastal erosion can alleviate this, with Arctic primary production (i.e. the turnover photosynthesising plankton biomass) projected to increase by about 30-50 per cent in this century. Invasive species can also extend further into the Arctic habitat due to warming and current changes, e.g. in the Barents Sea and from the Pacific (Kelly et al., 2020; Neukermans et al., 2018; Oziel et al., 2020; Terhaar et al., 2021).

In the North Atlantic, the AMOC can be defined as north-going warm 'limb' and saline upper waters and south-going, colder, denser deep water 'limb' (Frajka-Williams et al. 2019). In contrast, in the Subpolar North Atlantic and the SPG, the AMOC features a third 'limb' of a cold, fresh western boundary current with the origin in the Arctic Ocean and Nordic Seas (Bacon et al., 2022). This is likely linked with the deep convection and winter oceanic mixing in the Labrador, Irminger and Iceland seas, injecting waters into the deep, southward-flowing limb of the AMOC (Bower et al. 2019). Changes in SPG circulation are associated with the shallowing of the oceanic mixed layer and convection (Figure 6a,b) in the SPG and link the predicted future weakening of the North Atlantic subtropical gyre and a strengthening of the Nordic Seas gyre, pointing to the influences of the upstream changes in the Arctic on the North Atlantic (Swingedouw et al., 2021).

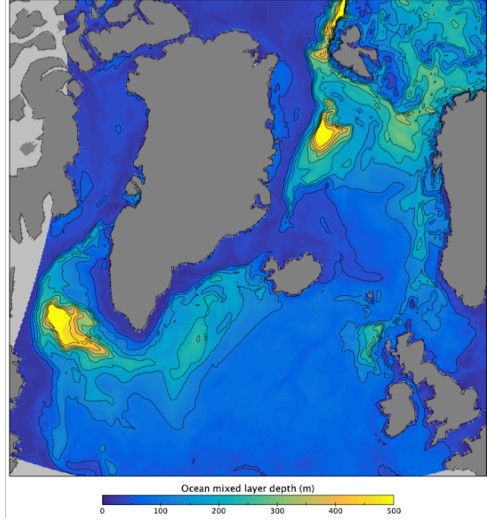

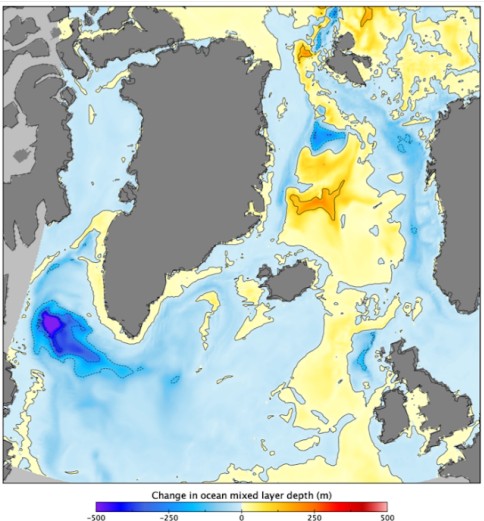

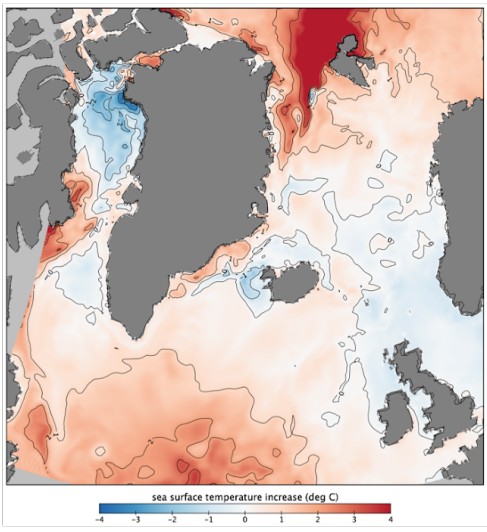

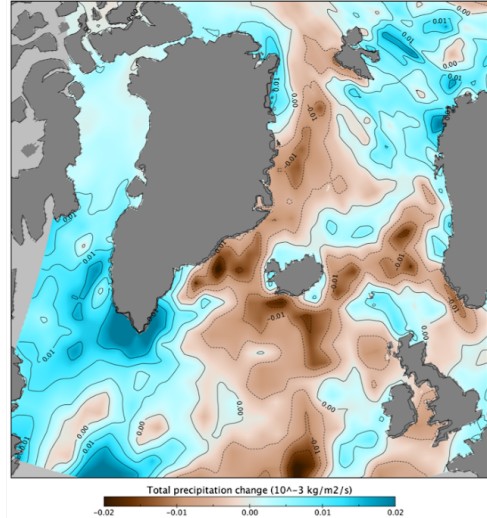

**Figure 6: Projected SPG changes. a) Winter ocean mixed layer depth (MLD) as indicator of ocean convection in winter 2020-30 (January-March). b) Changes in projected MLD by winter 2040-50 with respect to the 2020-30 baseline. c) Change in summer sea surface temperature (SST) and d) winter total atmospheric precipitation, respectively, projected by winter 2040-50. NEMO-MEDUSA 1/4 degree high resolution model results using ssp370 CMIP6 scenario 2015-2099. High-resolution simulations are courtesy of A. Coward, A. Yool, K. Popova and S. Kelly, NOC. Please see Supplementary Information for more analysis, and Swingedouw et al. (2021) for the IPCC CMIP6 model results.**

## 2.2.1 Evidence for tipping dynamics

Potential convection instability in the Labrador and Irminger Seas and the wider SPG is believed to be linked to lightening of the upper ocean waters due to reduced salinity (e,g., due to increased precipitation, Figure 6d), thus increasing 'stratification' – i.e. reduced mixing between layers of the water column. Warming (Figure 6c) also plays a role and could contribute to convection collapse (Armstrong McKay et al., 2022). Freshening and warming make surface waters more buoyant and thus harder to sink, which, beyond a threshold, can abruptly propel a self-sustained convection collapse (Drijfhout et al. 2015, Sgubin et al., 2017). This process can result in two alternative stable SPG states (Levermann and Born, 2007), with or without deep convection (Armstrong McKay et al., 2022). Similar to the AMOC, SPG stability is also strongly linked to the salt-advection feedback. When the SPG is 'on', it brings dense salty waters from the North Atlantic drift into the Irminger and Labrador Seas, allowing deep sinking and convection to occur (Born & Stocker, 2014; Born et al., 2016). When convection decreases due to stratification, the SPG weakens, fresher North Atlantic waters flows eastwards, and the convection is further weakened, which eventually leads to convection collapse in some models. SPG collapse leads to cooling across the SPG region, and so will impact marine biology and bordering regions.

A freshwater anomaly is currently building up in the Beaufort gyre – a pile-up of fresh water at the surface of the Beaufort Sea in the Arctic – due to increased freshwater input from rivers, sea ice and snow melting as well as the prevailing clockwise (anticyclonic in the northern hemisphere) winds over the sea (Haine, et al.. 2015; Reagan et al., 2019; Kelly et al., 2020). There is a considerable risk that this freshwater excess might flush into the SPG, disrupting the AMOC (Zhang and Thomas, 2021). The most recent changes in Beaufort gyre size and circulation (Lin et al. 2023) suggest flushing might occur very soon or has already started. The SPG system has recently experienced its largest freshening for the last 120 years in its eastern side due to changes in the atmospheric circulation (Holliday et al. 2020). In contrast, so far there is only limited evidence of Arctic freshwater fluxes impacting freshwater accumulation in the Labrador Sea (Florindo-Lopez et al., 2021). An increased freshwater input into SPG water mass formation regions from melting of Greenland's glaciers can also inhibit deep water formation and reduce the SPG and AMOC (Dukhovskoy et al., 2021).

Although SPG changes are apparently linked to the AMOC, the SPG collapse can occur much faster than AMOC collapse, on the timescale of only a few decades (Armstrong McKay et al., 2022), and independently (Sgubin et al., 2017). Armstrong McKay et al. (2022) estimated global warming threshold of ~1.8°C (1.1 to 3.8°C) for the SPG collapse (high confidence) based on climate models from CMIP5 and CMIP6. Abrupt future SPG collapse is diverse in the CMIP6 models, occurring as early as the 2040s (~1 to 2°C) but in only a subset of models. However, as these models better represent some key processes, the chance of SPG collapse is estimated at 36-44 per cent (Sgubin et al., 2017; Swingedouw et al., 2021).

## 2.2.2 Assessment and knowledge gaps

Similar to Armstrong McKay et al. (2022), the SPG is classified as a tipping system with medium confidence. A global warming threshold for tipping that could be passed within the next few decades, and an estimated tipping timescale of years

to a few decades, raise reasons for concern. Furthermore, cessation of deep water production from other sources in the
Labrador and Nordic Seas and the Arctic could also present other potential tipping points in the future North Atlantic
(Sgubin et al., 2017).

**2.3 Southern Ocean circulation**

Two main tipping points in the Southern Ocean have been discussed in the past, which both could have large and global
climate consequences. The first is the slowdown and collapse of the Antarctic Overturning Circulation; the second is the
abrupt change in ocean circulation on the Antarctic continental shelf, leading to suddenly rising ocean temperature in contact
with the Antarctic ice shelves fringing the ice sheet.

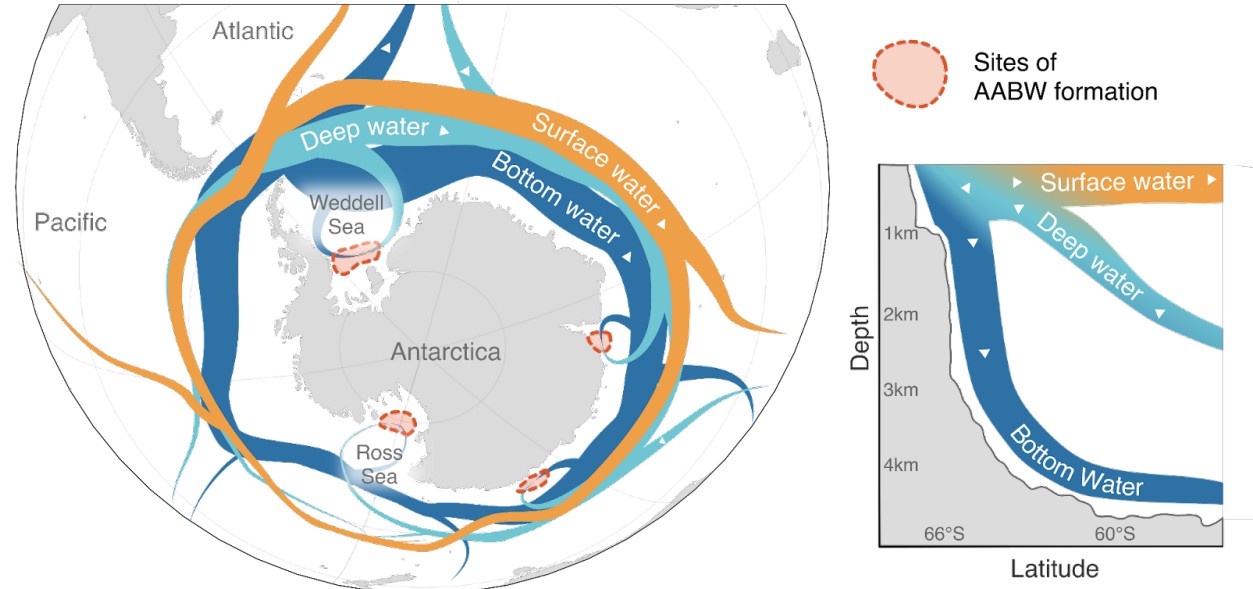

**Figure 7: Circulations and potential tipping systems in the Southern Ocean. Adapted from Li, England et al (2023) and IPCC SROCC Fig CB7.1**

Along with the AMOC, the Antarctic overturning circulation constitutes the second branch of the global ocean overturning circulation linking the surface to the abyssal ocean (Figure 7), forming Antarctic Bottom Water (AABW) through sinking of the shelf waters around the Antarctic continent. A key mechanism is brine rejection from sea-ice formation: very salty water that is left behind when ocean water freezes, which causes the ambient liquid water to become heavier and sink. This is maintained by offshore winds blowing away from the Antarctic continent, pushing sea ice away from the coast and forming areas of open water (so-called polynyas) supporting brine rejection. The formation of AABW sustains the operation of the lower branch of the Antarctic overturning circulation (Figure 7 and Abernathey et al., 2016).

In contrast to our understanding of the AMOC, any changes related to the future of the Antarctic Overturning Circulation has remained at low or medium confidence due to a persistent lack of process understanding (Fox-Kemper et al., 2021; Heuzé et al., 2021; Purich and England 2023). However, evidence of its ongoing decline has escalated in recent years, both from observations (Gunn et al., 2023; Zhou et al., 2023; including record low sea ice extent in 2022-2023) and numerical models (Lago & England, 2019; Liu et al. 2022; Li et al, 2023), linked to the changes in melt water, wind trends, sea ice transport and water mass formation (Holland et al., 2012).

Change or collapse in the Antarctic Overturning Circulation has the potential for widespread climate and ecosystem implications within this century. The Southern Ocean surface temperature is set by a delicate balance between ocean overturning strength, upper ocean stratification (the degree of mixing between ocean layers), air sea ice fluxes and sea ice cover. The Antarctic Overturning Circulation affects cloud feedbacks and has been shown to be a key regulator of Earth's

global energy balance, so much so that it is the main control on the timing at which the 2°C global warming threshold will be reached for a given emission scenario (Bronselaer et al., 2018, Dong et al., 2022; Shin et al., 2023).

Reduced Antarctic overturning can also shift global precipitation patterns, resulting in drying of the Southern Hemisphere and wetting of the Northern Hemisphere (Bronselaer et al., 2018). Reduced Antarctic overturning also reduces the efficiency of the global ocean carbon sink, leaving more nutrient-rich water at the seafloor (Liu et al., 2022), and also affects global ocean heat storage (Li et al., 2023). Amplifying feedbacks to further shelf water warming and ice melt are also possible (Bronselaer et al., 2018; Purich and England 2023; Li et al., 2023).

### 2.3.1 Evidence for tipping dynamics

Different generation climate models consistently project a slowing or collapse of the Antarctic overturning under a warming climate (Heuzé et al., 2015, 2021; Lago and England, 2019; Meredith et al., 2019; Fox-Kemper et al., 2021; Liu et al., 2022). However, our confidence in these models to assess change in Antarctic overturning is limited due to known limitations in the representation of dense water formation (Purich and England 2023). Limitations come also from the lack of representation of increased Antarctic ice sheet meltwater in most models (Fox-Kemper et al., 2021). Armstrong McKay et al. (2022) identified the Antarctic Overturning Circulation as a potential but uncertain tipping system in the climate system, but gaps in process understanding meant a threshold remained uncertain. They estimated the Antarctic Overturning Circulation to be prone to collapse at a global warming level of 1.75-3°C based on Lago & England (2019).

Specifically designed model experiments aiming to bridge some of these limitations, in combination with evidence from observed changes (Gunn et al., 2023; Purkey and Johnson, 2013), suggest that we are currently heading toward a decline and possible collapse of the Antarctic Overturning Circulation (Li et al., 2023, Zhou et al., 2023). The rapidity of this decline might even be underestimated, according to recent observations (Gunn et al., 2023). The sensitivity of the overturning circulation to increases in upper ocean stratification is also consistent with paleo evidence. Observations from marine sediments suggest that AABW formation was vulnerable to freshwater fluxes during past interglacials (Hayes et al., 2014; Huang et al., 2020; Turney et al., 2020) and that AABW formation was strongly reduced (Skinner et al., 2010; Gottschalk et al., 2016; Jaccard et al., 2016) or possibly totally curtailed (Huang et al., 2020) during the Last Glacial Maximum and earlier transient cold intervals.

Local water mass characteristics and associated circulation regimes on the Antarctic continental shelf are setting the rate of ice shelf melt rates in ice 'cavities', the regions of ocean water covered by floating ice shelves. Relatively warm water reaching the continental shelf in west Antarctica causes high basal melt rates with severe consequences for the ice shelf, ice sheet dynamics, and sea level rise (Naughten et al., 2023). In contrast, the largest ice shelf cavities in the Weddell and Ross Seas are not exposed to this relatively warm water, and consequently have melt rates orders of magnitude smaller than in West Antarctica. Despite this, the Weddell and Ross Sea ice shelf cavities have been shown to exhibit tipping behaviour (Hellmer et al., 2012; 2017; Siahaan et al. 2022). Models show that they are prone to sudden warming of their cavity under

future climate change, dramatically increasing basal melting with important consequences for global sea level rise (Hellmer et al., 2012; 2017; Siahaan et al. 2022). Once tipped into a warm state, such cavities could be irreversibly maintained in such a state, even when forcing is reduced (Hellmer et al., 2017). However, it remains unclear what threshold would need to be crossed to tip those cavities from a cold to warm state, and it may only occur under extreme climate change scenarios.

### 2.3.2 Assessment and knowledge gaps

In summary, the combination of process-based understanding and observational, modelling and paleoclimate evidence suggests that the Antarctic Overturning Circulation will continue to decline in the 21st Century. There is increasing evidence for positive amplifying feedback loops that can lead to a collapse of the overturning, with widespread global climate and ecosystem consequences. Closely linked to this is a potential tipping in continental shelf water temperature, driven by amplifying meltwater feedbacks once a regional temperature threshold is crossed. We therefore classify the Southern Ocean Circulation as a tipping system with medium confidence. However, its potential tipping thresholds remain uncertain.

# 3. Atmospheric Circulations: Monsoons

Monsoon circulations are large-scale seasonal changes in the direction and strength of prevailing winds over South Asia, East Asia, Africa, Australia and the Americas. Historically, monsoons were seen as large-scale sea breeze circulations driven by land-sea heating differences due to seasonal changes in incoming solar radiation (Figure 8). Currently, a perspective of a global monsoon has emerged (Trenberth et al., 2000; Wang & Ding, 2008), whereby the monsoon systems are seen as interconnected and driven by localised seasonal and more extreme migrations of the ITCZ (Gadgil, 2018; Geen et al., 2020, and references within). Monsoon regions in the world experience heavy precipitation in the summer months, and the global monsoon system is an integral part of the global hydrological cycle, contributing ~31 per cent of total precipitation over the globe (Wang and Ding, 2008).

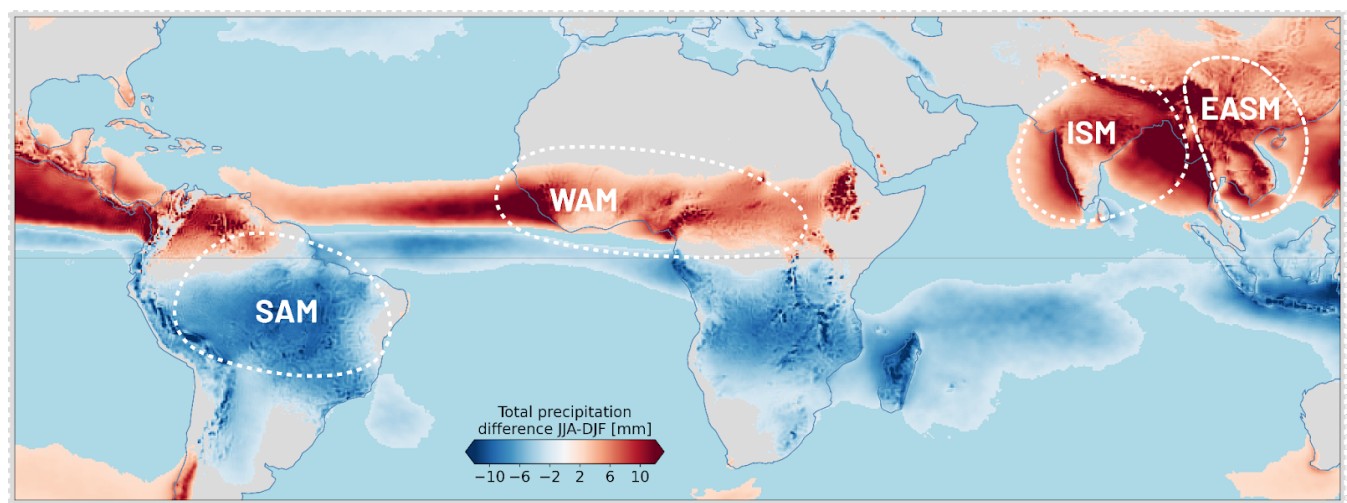

**Figure 8: Monsoon systems. Shown is the total precipitation difference between Northern hemisphere summer (June-August, JJA) and winter months (December-February, DJF), highlighting the dominant precipitation patterns over South America (SAM), West Africa (WAM), India (ISM) and East Asia (EASM). Based on data from ERA5 (Hersbach et al., 2023), with monthly averages over 1980-2010.**

There is a recent intensification trend in global monsoon precipitation, mainly due to enhanced northern hemisphere summer monsoon (Wang et al., 2012). It will likely continue in the future (high confidence, IPCC AR6, by ~1-3% per °C warming) because of increased water vapour related to warming driven by increased $CO_2$ in the atmosphere (Hsu et al., 2013; Lee and Wang, 2014; Chen et al., 2020; Moon and Ha, 2020; Wang et al., 2020), although a few studies conversely show that climate warming may lead to a weakened global monsoon circulation (Hsu et al., 2012, 2013). Climate simulations also project expansion of global monsoon domain areas with increasing $CO_2$ (Wang et al., 2020; Paik et al., 2023) and increased frequency of monsoon precipitation extremes in the 21st Century (Chevuturi et al., 2018; Ali et al., 2020, Modi and Mishra, 2019; Moon and Ha, 2020; Katzenberger et al., 2021).

Monsoon precipitation is vital for agrarian populations and livelihoods in vast areas of South Asia, Africa, and South America, and changes to it could expose almost two thirds of the global population to disastrous effects (Wang et al., 2021). Hence it is crucial to understand the dynamics and potential nonlinear changes or tipping behaviour of monsoon systems under a changing climate. Here the 'tipping' of monsoon systems refers to a significant, feedback-driven shift in the precipitation state of the monsoon, with implications for the regional and global climate and ecosystems. In this discussion we assess if monsoon systems show any evidence of nonlinear (tipping or abrupt) responses to climate forcings based on available literature.

## 3.1 Indian summer monsoon (ISM)

During the summer season over South Asia (June-September), winds from the south west carry large amounts of water vapour from the Indian Ocean to the Indian subcontinent and cause heavy precipitation in the region, providing ~80 per cent of the total annual precipitation (Figure 8). ISM precipitation shows considerable intra-seasonal, interannual, and decadal variability, many times with precipitation extremes (leading to droughts, floods) during the season, and years and decades with above and below (in drought years) normal precipitation. Indian monsoon variability is strongly influenced by ocean-atmosphere interactions such as El Niño Southern Oscillation (ENSO), Indian Ocean Dipole events (irregular changes in the temperature gradients in the Indian Ocean, Cherchi et al., 2021; Chowdary et al., 2021; Hrudya et al., 2021), and cooler temperatures in the North Atlantic (Borah et al., 2020).

ISM precipitation declined in the second half of the 20th Century, attributed mainly to human-driven aerosol loading (Bollasina et al., 2011) and strong Indian Ocean warming (Roxy et al., 2015). Recent studies (Jin and Wang, 2017) suggest the ISM has revived since 2002, linked to enhanced warming over the Indian subcontinent due to reduced low clouds, resulting in an increased land-ocean thermal gradient. Future projections suggest increases in the ISM precipitation in future warming scenarios (by 5.3 per cent per celsius of global warming according to CMIP6 models, Katzenberger et al., 2021) and a longer monsoon duration (Moon and Ha, 2020).

## 3.1.1 Evidence for tipping dynamics

Many periods of abrupt ISM transitions have been identified in past monsoon records in association with high-latitude climate events (Schulz et al., 1998; Morrill et al., 2003) such as during Heinrich events (glacial outbursts that temporarily shut down the AMOC) (McManus et al., 2004; Stager et al., 2011), the Younger Dryas (a temporary return to more intense glacial conditions between 12,900–11,700 years ago, Carlson et al., 2013; Cai et al., 2008), and several periods during the more recent Holocene (Gupta et al., 2003; Berkelhammer et al., 2012; Yan and Liu, 2019). However, the mechanisms of such abrupt transitions are not clearly understood. Efforts have been made to identify any Indian monsoon tipping mechanisms using simplified models (Zickfeld et. al., 2005; Levermann et al., 2009).

An internal feedback mechanism, a "positive moisture advection feedback" (Zickfeld et. al., 2005; Levermann et al., 2009; Schewe et al., 2012), has been suggested as responsible for abrupt transitions simulated using these analytical models. In this feedback, the atmospheric temperature gradient between the land and cooler ocean in summer leads to the onshore transport of moist air (advection), which then rises, forms clouds, and condenses into rain. The phase transition from vapour to liquid warms the surrounding air (through the release of latent heat, or 'diabatic heating'), increasing the land-ocean temperature gradient, and sustaining this monsoon circulation. Any forcing that weakens this pressure gradient can therefore lead to monsoon destabilisation (Zickfeld et al., 2005). If monsoon winds weaken, advection and condensation reduce, and the threshold for a monsoon tipping is reached when the diabatic heating fails to balance the heat advection away from the region (Levermann et al., 2009).

Contrarily, follow-up studies (Boos and Storelvmo, 2016) challenge occurrence of any tipping in these simplified models, and rule out any abrupt monsoon responses to human-driven forcings in the future, and instead attribute past monsoon shifts to rapid forcings or vegetation feedbacks. Simplified models omit key aspects and feedbacks in the monsoon system (specifically, static stability of the troposphere in the models that simulated the monsoon tipping, Boos and Storelvmo, 2016; Kumar and Seshadri, 2023). Hence, more studies using models that represent the complexities of the monsoon and paleoclimate data are required for a clearer picture on any non-linear changes in the monsoon system.

Apart from climate change, aerosols pose another significant human-driven pressure on the Earth system. Aerosols influence the Earth's radiative budget, climate, and hydrological cycle by reflecting or absorbing solar radiation, changing the optical properties of clouds, and also by acting as cloud condensation nuclei. An increase in anthropogenic aerosols has been attributed as the major reason for the decline of Northern Hemispheric summer monsoon strength from the 1950s to 1980s (Cao et al., 2022), due to its dimming effect.

A large increase in regional aerosol loading over South and East Asia (>0.25 Aerosol Optical Depth, AOD, Steffen et al., 2015) could potentially switch the Asian regional monsoon systems to a drier state. Further, hemispheric asymmetries in the aerosol loading (>0.15 AOD Rockström et al., 2023), due to volcanic eruptions, human sources, or intentional geoengineering, could lead to hemispheric temperature asymmetries and changes in the location of the ITCZ, significantly disrupting regional monsoons over West Africa and South Asia (Haywood et al., 2013; Rockström et al., 2023; Richardson et al., 2023). However, there is no direct evidence of aerosols causing a tipping of the monsoon systems, and uncertainties in threshold estimates are large due to complex aerosol microphysics and aerosol-cloud interactions. Hence, systematic observational and modelling approaches would be needed to reduce the uncertainties, as well as additional assessments of interhemispheric asymmetries in the aerosol distribution.

**3.1.2 Assessment and knowledge gaps**

The Indian summer monsoon system was earlier classified as one of the Earth's tipping systems (Lenton et al., 2008), based on the threshold behaviour of the ISM in the past and the moisture-advection feedback (Levermann et al., 2009), which was refuted by later studies (Boos and Storelvmo, 2016; Seshadri, 2017). Most recently, Armstrong McKay et al. (2022) categorise ISM as an "uncertain potential [climate] tipping element" as global warming is not likely to cause tipping behaviour directly in ISM precipitation.

Based on this current literature, the chances for ISM exhibiting a tipping behaviour towards a new low-precipitation state under climate change are uncertain, warranting extensive studies on the subject. However, potential tipping behaviour in the AMOC (relation to global monsoon described in West African monsoon below) or increase in the interhemispheric asymmetry of aerosol loading in the atmosphere beyond potential threshold levels could lead to large disruptions to monsoon systems. This could cause calamitous effects on millions of people in the monsoon regions, even in the absence of tipping.

## 3.2 West African monsoon (WAM)

The West African monsoon (WAM) controls hydroclimatic conditions, vegetation, and mineral-dust emissions of northern tropical and subtropical Africa, up to the semi-arid Sahel region at the southern edge of the Sahara Desert (Figure 8). The strength of the monsoon shows large variations over a range of timescales from interannual to decadal and longer. Albedo (reflectivity of the Earth's surface) changes caused by human-driven land-cover changes and desertification (Charney et al., 1975; Charney, 1975; Otterman, 1974) can affect rainfall: a less vegetated surface with higher albedo increases radiative loss, thereby reducing temperature and suppressing the rising and condensation of moist air into rainfall (i.e. convective precipitation). Variations of sea surface temperatures (SSTs) in different oceanic basins can also drive interannual and decadal variability in WAM precipitation (Rodríguez-Fonseca et al., 2015). Other major factors that affect WAM variability are land surface variability such as variations in soil moisture (Giannini et al., 2013; Zeng et al., 1999), vegetation (Charney et al., 1975; Kucharski et al., 2013; Otterman, 1974; Wang et al., 2004; Xue, 1997), high-latitude cooling (Collins et al., 2017), and dust emissions (Konare et al., 2008; Solmon et al., 2008; Zhao et al., 2011). Charney et al. (1975) proposed the vegetation-climate positive feedback mechanism to explain the decadal precipitation variability in the WAM. The feedback mechanism is explained as desertification leads to increased albedo, surface cooling, and increased stability in the atmosphere, hence, reduced convection and precipitation, thereby amplifying the initial response. It has since been suggested that additional factors such as evapotranspiration, dust emissions, and water vapour recycling related to the vegetation further contribute to the vegetation-climate feedback (Rachmayani et al., 2015; Pausata et al., 2016; Yu et al., 2017; Chandan and Peltier, 2020).

## 3.2.1 Evidence for tipping dynamics

Paleoclimate records underscore dramatic variations of the WAM in the more distant past, such as the periodic expansion of vegetation into the Sahara Desert during the so-called 'African humid periods' (AHPs) and are linked to the emergence of ancient cultures along the Nile. The vegetation-climate feedback likely played a significant role in amplifying and sustaining the wetter conditions and facilitating a green Sahara during AHPs (Pausata et al., 2016). Another example is the drought 200–300 years ago, which caused the water level of Lake Bosumtwi in Ghana to fall by almost four times as much as it did during the drought of the 1970s and 1980s. Large past variations of the WAM, such as those during the AHPs, raise the question of whether present day anthropogenic global warming could have potentially significant impacts on the WAM. Although the nature and magnitude of radiative forcing were different during the AHPs than they are now (i.e. an external change in insolation due to orbital forcing versus an internal change from increased greenhouse gases), the fact that the AHPs occurred under a globally warmer climate than the pre-industrial period invites questions.

Some paleoclimate archives show WAM precipitation changes that took place over several centuries (deMenocal et al., 2000; McGee et al., 2013), i.e. an order of magnitude faster than the orbital forcing. However, others show a much more gradual change (e.g. Kröpelin et al. 2008) with a time-varying withdrawal of the WAM from North to South following the

insolation changes (Shanahan et al., 2015). Because of geographic variability of the African landscape and African monsoon circulation, abrupt changes can occur in several, but not all, regions at different times during the transition from the humid to arid climate (Dallmeyer et al., 2021).

By inducing latitudinal movements of the ITCZ, change in the Atlantic Meridional Overturning Circulation (AMOC) is considered to play a role in shifts of global monsoon systems (Stager et al., 2011). Paleoclimate evidence suggests that glacial meltwater-induced weakening of the AMOC during Heinrich events in the last glacial period led to abrupt Asian and African monsoon weakening (Mohtadi et al., 2014; 2016). Similarly, the Younger Dryas led to a cool and dry state over Northern Hemisphere tropical monsoon regions, as well as the 8.2-kyr event during the Holocene (Alley and Ágústsdóttir, 2005). North Atlantic fresh water-hosing simulations using climate models (Lewis et al., 2010; Pausata et al., 2011; Kageyama et al., 2013) confirm these shifts in ITCZ can occur as a result of substantial glacial meltwater release. These influences of AMOC on the monsoon systems have also been studied in the context of the South American monsoon (see below). Hence, a collapse of AMOC has the potential to cause disruptions to the regional monsoon systems and other tropical precipitation systems over Asia, Africa, and South America (Gupta et al., 2003; IPCCAR6; Ben-Yami et al., 2024). Additionally, location and intensity of the tropospheric jet streams have been identified as processes modulating the strength and position of the African rainbelt (e.g., Farnsworth et al., 2011, Nicholson and Dezfuli, 2013).

### 3.2.2 Assessment and knowledge gaps

Abrupt changes in one region can be induced by abrupt changes in other regions, a process sometimes referred to as "induced tipping" (Pausata et al., 2020). The AHP transition of the Sahara was slow with respect to timescales of individual humans and local ecosystems, but regionally rapid with respect to changes in the driver. Based on the record of large past variations of WAM precipitation patterns (including collapse), and the existence of positive amplifying feedbacks, we classify WAM as a tipping system with low confidence. This is in line with previous assessments (Armstrong McKay et al., 2022), in which a lower tipping threshold of 2°C global warming was estimated but attributed low confidence due to limited model resolution of vegetation shifts, and model disagreements in future trends. The timescale of abrupt shifts is estimated to range from decades as observed in CMIP5 models (Drijfhout et al., 2015) to centuries based on paleorecords (Hopcroft and Valdes, 2021; Shanahan et al.; 2015). Potential additional destabilisation through AMOC weakening and atmospheric aerosol loading, and the far-reaching implications of WAM tipping call for intensified research efforts on this system.

### 3.3 South American Monsoon (SAM)

The South American Monsoon (SAM) system is characterised by strong seasonality in precipitation, even though it does not show a reversal of low-level winds like the Asian monsoon (Zhou and Lau, 1998; Vera et al., 2006; Liebmann and Mechoso, 2011; Carvalho et al., 2012). Studies on the SAM system are relatively few compared to the Asian and African monsoon systems, as it was not classified as a monsoon system until a couple of decades ago (Zhou and Lau, 1998).

A mature SAM system (from December to February) shows features such as enhanced northeastern trade winds, increased land-ocean thermal gradient, and the development of an active convective zone (the South Atlantic Convergence Zone) (Figure 8; Zhou and Lau, 1998). The SAM system affects vast areas of tropical South America all the way to southern Brazil, and provides more than 50 % of the annual precipitation to these regions (Vera et al., 2006) including most of the Amazon rainforest. SAM precipitation varies from interannual to orbital timescales (Chiessi et al., 2009; Liebmann and Mechoso, 2011; Carvalho and Cavalcanti, 2016; Hou et al., 2020).

The influence of anthropogenic climate change on the SAM precipitation is ambiguous (Douville et al., 2021), and many CMIP5/CMIP6 models are noted for their poor representation of SAM precipitation (Jones and Carvalho, 2013; Douville et al., 2021). IPCC AR6 finds high confidence in delayed onset of the SAM precipitation since the 1970s associated with climate change, which could worsen with increased $CO_2$ levels (Douville et al., 2021). However, the projected future change in total SAM precipitation is uncertain, as the models show low agreement on the projections (Douville et al., 2021).

### 3.3.1 Evidence for tipping dynamics

Orbital timescale changes (i.e. over 10s of 1000s of years) in SAM precipitation seem to be largely controlled by changes in insolation and respond linearly to it (Cruz et al., 2005; Hou et al., 2020). Millennial-scale changes (i.e. over 1000s years) in the SAM are thought to be associated with variations in the strength of the AMOC as described for the West African monsoon (see above). In particular, paleo evidence indicates that an increase in tropical South American precipitation to the south of the equator followed a weakening of the AMOC related to Heinrich stadials (Mulitza et al., 2017; Zhang et al., 2017; Campos et al., 2019). Similarly, meltwater flux from the Laurentide Ice Sheet during the Younger Dryas probably led to wet anomalies over tropical South America to the south of the equator (McManus et al., 2004; Broecker et al., 2010; Venancio et al., 2020; Brovkin et al., 2021). Earth system model projections of AMOC collapse impacts on tropical South American rainfall are model dependent, but generally find a reduction in rainfall over northernmost South America and an increase over tropical South America to the south of the equator (Bellomo et al., 2023; Nian et al., 2023; Orihuela-Pinto et al., 2022a; Liu et al., 2020).

Deforestation of the Amazon rainforest may force a tipping dynamic in the SAM system. In a non-linear model for moisture transport, deforestation over 30-50 % of the Amazon rainforest led to a tipping point in the SAM system, causing precipitation reductions of up to 40 % in non-forested parts of the western Amazon (Boers et al., 2017). This reduction in SAM precipitation is caused by the breakdown of a positive amplifying feedback mechanism present in Amazonian precipitation that involves latent heat of condensation over the Amazon rainforest due to transpiration (i.e. water lost from plants) and water vapour transport from the Atlantic. Reduced transpiration due to deforestation can no longer sufficiently provide water vapour to sustain the latent heat required, thereby reducing the inflow of oceanic water vapour, and leading to a SAM tipping in this model (Boers et al., 2017). In addition to deforestation, transpiration over northern Amazon may be reduced due to an increase in seasonal tropical vegetation related to an AMOC slowdown (Akabane et al., 2024), further curtailing moisture availability for the SAM.

### 3.3.2 Assessment and knowledge gaps

A combination of climate change and deforestation could lead to substantial changes in the SAM system, affecting many millions of people. Additionally, a decrease in AMOC strength could potentially trigger major changes in tropical South American precipitation (see Wunderling et al., 2023). However, the current scarcity of research in the subject limits our ability to fully understand and assess the tipping potential of the SAM system, and we classify the possibility of SAM tipping to be uncertain.

### 3.4 East Asian Summer Monsoon (EASM)

Beyond the major tropical monsoon systems, there are other subtropical monsoon systems which ostensibly share similar characteristics, such as the East Asian Summer Monsoon (EASM), which is a part of the South Asian Monsoon. Unlike the tropical monsoon systems which are driven by the ITCZ movement and characterised by heavy summer rainfall, the precipitation and wind patterns of the EASM are linked to mid-latitude frontal systems and the jet stream, and its interaction with the Tibetan Plateau (Molnar et al., 2010; Son et al., 2019). A recent study on CMIP6 projections of EASM by Katzenberger and Levermann (2024) finds that the future precipitation and interannual variability in the EASM region is projected to consistently increase under various climate change scenarios. They suggest these changes are primarily driven by thermodynamic responses to global warming, rather than major shifts in atmospheric circulation patterns. Further, the same study finds an increased frequency of extreme wet seasons based on the CMIP6 simulations.

### 3.4.1 Evidence for tipping dynamics

Speleothem and loess records from China, and analyses based on these records (Wang et al., 2008; Thomas et al., 2015; Rousseau et al., 2023) reveal that the EASM has undergone periods of abrupt changes on millennial timescales. These changes are more sudden than the change in Northern Hemisphere summer insolation that is thought to be the primary driver, and this suggests a potential tipping behaviour in the EASM. Schewe et al. (2012) proposed a mechanism involving moisture advection feedback to explain these abrupt changes in the past, and suggests that a minimum threshold of specific humidity over the ocean is essential to sustain monsoon precipitation over the land. The changes in insolation may trigger changes in sea-surface temperature and evaporation, hence specific humidity over the ocean. According to the conceptual model proposed by Schewe et al. (2012), gradual, insolation-driven changes in oceanic humidity lead to abrupt transitions-the system suddenly switches between states that can or cannot sustain a strong monsoon circulation whenever the oceanic specific humidity crosses the threshold. Although paleoclimate records show evidence of past abrupt shifts in the EASM, the mechanism detailed in Schewe et al. (2012) was contested by a later study by Boos and Storelvmo (2016). They highlight that the underlying theory in Schewe et al. (2012) omitted the static stability of the troposphere. Modeling experiments by Boos and Storelvmo (2016), primarily examining decadal timescales, find a more linear monsoon response to forcings. They

suggest that the mechanisms behind orbital-scale abrupt changes might involve very large forcings or slow-acting components like ice sheets and large-scale vegetation changes.

Further, like the Indian, African, and South American monsoons, the EASM could be affected by a potential collapse of the AMOC (Sun et al., 2012; Ben-Yami et al., 2024). Freshwater hosing experiments using climate models that mimic AMOC slowdown show increased temperature gradients between the equator and the poles, and suggest northern westerlies acting as a pathway in transmitting climate change signals from the North Atlantic to the Asian monsoon regions (Sun et al., 2012).

### 3.4.2 Assessment and knowledge gaps

While paleoclimate records provide compelling evidence for EASM's tipping behaviour in the past, a deeper understanding of the underlying mechanisms and internal feedback is crucial for assessing its vulnerability to future tipping points. We therefore classify this system as uncertain.

### 3.5 Summary

The major monsoon systems discussed here share some similar, underlying dynamics, as they are complex phenomena emerging from the interplay of global and regional factors. As such, questions around potential tipping largely relate to the susceptibility of changes in the global monsoon to anthropogenic forcing, e.g. either directly via interhemispheric differences in aerosol loading or indirectly via climate-change driven AMOC collapse. In principle, regional feedbacks may maintain or significantly shift a monsoon regime but there is insufficient evidence across the discussed systems, with exception of the West African monsoon. This system does not only feature strong positive feedbacks involving vegetation and dust, but has also reportedly collapsed abruptly several times in the past. Overall, we therefore classify tipping of the West African monsoon as possible with low confidence, and South American, East Asian and Indian monsoons as uncertain.

# 4. Global atmospheric circulations

## 4.1 Tropical clouds, circulation, and climate sensitivity

Clouds play an important role in the climate system, as they contribute to the regulation of Earth's energy budget. In general, the altitude and composition of clouds determines the amount of shortwave radiation to reach the planet's surface, as well as the longwave radiation (heat) escaping to space. For example, low, thick clouds have a net cooling effect as they deflect large fractions of incoming sunlight, while also having a weak greenhouse warming effect due to their low altitude (i.e. having a comparable temperature to the ground). A changing climate, which causes changes in temperature, humidity, and circulation patterns, affects the formation and dynamics of clouds. This, in turn, can influence the climate and how much warming results from increased atmospheric $CO_2$ concentrations (i.e. 'climate sensitivity').

### 4.1.1 Evidence for tipping dynamics

Literature on cloud-induced tipping points is very limited. Yet cloud-forming processes exhibit strong hysteresis on weather timescales. Indeed, a cloud droplet forms when water starts to stick to a particle after a certain level of humidity (in which a so-called hygroscopic aerosol particle crosses a humidity tipping point into an unstable condensational growth phase); and precipitation, once initiated, is a self-reinforcing cascade where larger particles fall faster and hence grow faster by collisions. Coupling of these micro-scale processes to atmospheric dynamics can lead to spontaneous and irreversible transitions at the intermediate mesoscale – in particular, the transition of shallow cloud layers from closed to open-cell geometries (honeycomb-like cloud patterns formed by convecting air) (Feingold et al. 2015), driven by a self-sustaining process involving precipitation-induced downdraft, subsequent updraft and new cloud formation (Yamaguchi and Feingold, 2015). Another example is self-aggregation of deep convection, apparently driven by radiative and moisture feedback loops (Müller et al. 2022). Both of these significantly decrease cloud cover and albedo, potentially enabling climate interactions. Could further coupling out to planetary scales produce climate-relevant tipping behaviour? Complicating this question is the fact that cloud-related processes are not well represented in current climate models, limiting their ability to guide us.

The most discussed possibility has been the extreme case of a global climate runaway. If the atmosphere became sufficiently opaque to infrared (i.e. if it became harder for longwave heat energy to escape due to overcast high cloud, very high humidity, or CFC-like greenhouse gases filling in spectral absorption windows), the planet could effectively lose its ability to cool to space, producing a Venus-like runaway. Although general circulation models (GCMs) and paleoclimate evidence suggest climate sensitivity rises as climate warms (Sherwood et al., 2020), calculations show virtually no chance of runaway warming on Earth at current insolation levels (LeConte et al. 2013).

A more plausible scenario is unexpectedly strong global positive amplifying radiative feedback from clouds and high climate sensitivity. Although presumably reversible, this would be serious. With respect to high clouds, suggested missing feedbacks (due to novel microphysical or aggregation mechanisms) have generally been negative (e.g. Mauritsen and Stevens 2015). Low clouds are a greater concern: one recent study using a multiscale atmospheric model found a strong and growing positive amplifying feedback from rapid disappearance of these clouds (Schneider et al. 2019), highlighting the possibility of nonlinear cloud behaviour and surprises (Bloch-Johnson et al.. 2015, Caballero and Huber 2013). Although various observations generally weigh against high-end climate sensitivities above 4oC per CO2 doubling, they cannot rule them out (Sherwood et al., 2020).

A final possibility is surprising reorganisations of tropospheric circulation. Innovative atmospheric models (Carlson and Caballero 2016, Seeley and Wordsworth 2021) and geologic evidence (Tziperman and Farrell, 2009, Caballero and Huber 2010) have suggested possible "super-MJO" (the 'Madden-Julian Oscillation' being the dominant mode of 'intraseasonal' variability in the tropical Indo-Pacific, and is characterised by the eastward spread of enhanced or suppressed tropical rainfall lasting less than a season) and/or reorganisation of the tropical atmospheric circulation in a warmer climate due to

cloud-circulation coupling. These scenarios are supported by little evidence, but if they did occur they could massively alter hydrology in many regions. Poor representation of tropical low clouds has also likely inhibited coupled model simulations of decadal variability or regional trends (Bellomo et al. 2014, Myers et al. 2018), raising the possibility that even if clouds cannot drive tipping points they might amplify other tipping points in ways that are missing from current models.

### 4.1.2 Assessment and knowledge gaps

In summary, concern about cloud-driven tipping points is relatively low. Cloud feedbacks will however likely affect the strength of climate responses, including for many tipping points. For example, they could potentially amplify variability, and current models may not be capturing this well. High climate sensitivity from strongly positive cloud feedbacks also cannot be ruled out.

### 4.2 El Niño-Southern Oscillation (ENSO)

The El Niño-Southern Oscillation (ENSO) is the dominant interannual mode of variability in Earth's climate. It originates in the tropical Pacific, where it affects sea surface temperatures (SST), trade winds, rainfall, and many other climate variables. El Niño events typically happen every 3 to 5 years (hence the term 'interannual'). The tropical Pacific average climate is characterised by a strong east-west gradient along the equator of about 5-6ºC, with warmer SSTs in the west and colder SSTs in the east maintained by easterly Pacific trade winds. During El Niño, the warm phase of this oscillation, this gradient weakens, while during La Niña, its cold phase, it intensifies (examples are shown in Figure 9a). Both phases of this oscillation have far-reaching impacts on global climate and weather patterns, ecosystems, and human health (e.g. McPhaden et al. 2020).

The impacts of ENSO become especially pronounced during the strongest events, often referred to as extreme El Niños (Figure 9b). At their peak, these events can eliminate the east-west ocean temperature gradient along the equator, leading to a temporary collapse of the trade winds. Additionally, an extreme El Niño causes an increase in global mean surface temperature of up to 0.25 °C (e.g. Hu and Fedorov 2017), contributing to the prevalence of heat waves around the globe. While only a few El Niño events reach large magnitudes, the global impacts of these events result in billions of dollars in damage (e.g. Callahan and Mankin 2023).

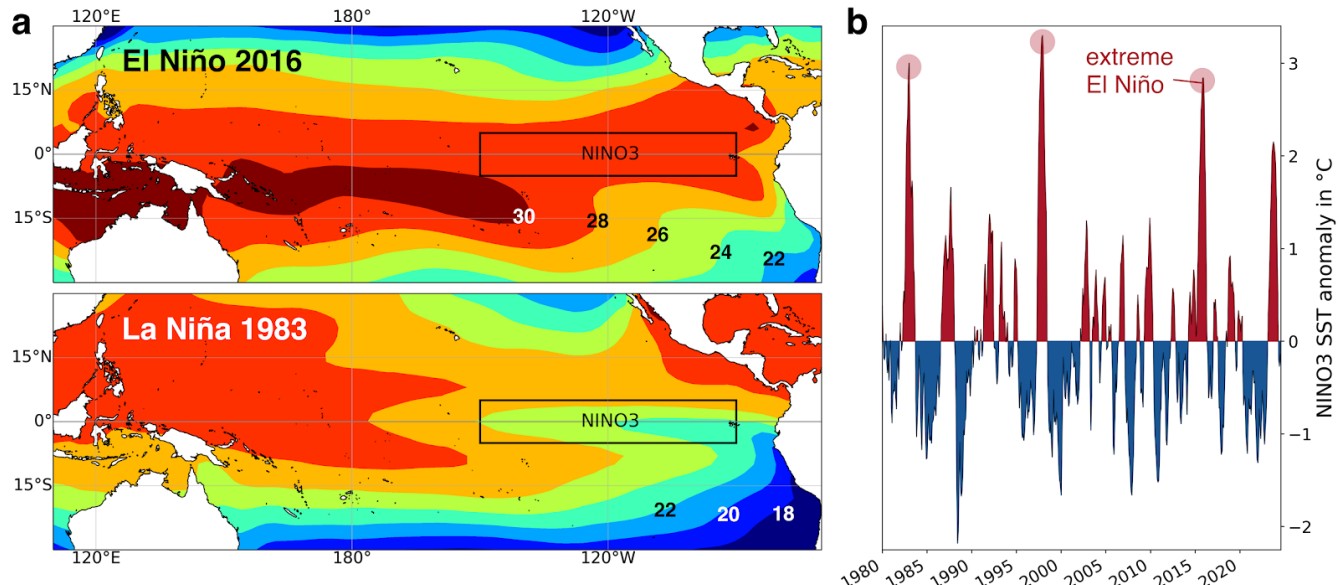

**Figure 9: ENSO warm and cold phases and observational record. a Examples of strong La Niña (top) and El Niño (bottom) events seen in the tropical Pacific surface temperature (SST) distribution, with characteristic strong and weak SST gradient along the equator, respectively. b ENSO record since the 1980s. Observed SST variations in the eastern equatorial Pacific since 1980. Note the three extreme events of the past four decades (1982, 1997 and 2015) and the weakening of ENSO variability between years 2000 and 2015. Temperature is averaged for the NINO3 region (5ºC-5ºN, 150ºW-90ºW) in the eastern equatorial Pacific. Based on NOAA Extended Reconstructed SST V5 data (Huang et al., 2017).**

### 4.2.1 Evidence for tipping dynamics

Extensive research conducted since the 1980s has significantly advanced our understanding of the physics behind El Niño, leading to improved predictive capabilities of climate models (L'Heureux et al., 2017). ENSO is now recognised as a large-scale, irregular, internal oscillatory mode of variability within the tropical climate system influenced by atmospheric noise (Timmermann et al., 2018). The spatial pattern of ENSO is determined by ocean-atmosphere feedbacks, while its timescale is determined by ocean dynamics. In particular, it is a sequence of self-reinforcing feedbacks between SSTs, changes in zonal surface winds, equatorial upwelling, zonal equatorial currents, and ocean thermocline depth that promotes the growth of El Niño anomalies and contributes to the positive Bjerknes feedback (McPhaden et al. 2020) .

Coral-based proxy data indicate that the amplitude and frequency of ENSO events has gradually increased during the Holocene (Grothe et al., 2019; Lawman et al., 2022), possibly due to an increase in strong El Niño events. All extreme El Niños in the observational record (1982, 1997, and 2015) occurred during the accelerated growth of global mean temperatures. This raises the question whether this trend is indicative of upcoming changes in the tropical Pacific to conditions with more frequent extreme El Niño events.

In the context of tipping points, the question arises: is there a critical threshold with an abrupt and/or irreversible transition
to such a new state? While there are no indications so far for a tipping point in ENSO behavior, several recent studies (e.g.
Cai et al. 2018, 2022; Heede and Fedorov 2023a) have indeed suggested that El Niño magnitude and impacts may intensify
under global warming (Figure 10), even though there is still no model consensus on the systematic future change in ENSO
as IPCC AR6 and the results in Figure 10 suggest.

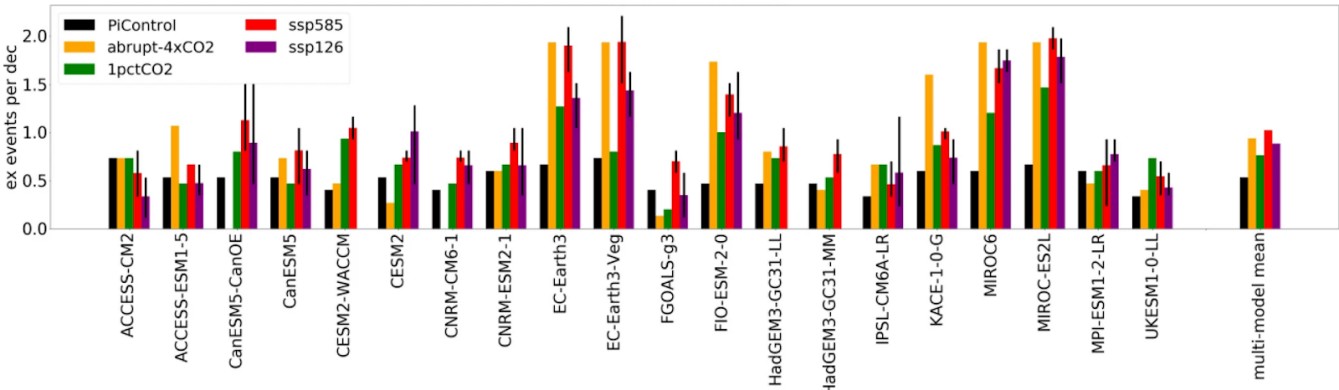

**Figure 10: Overview of projected changes in extreme El Niño events in CMIP6 climate models. The bar chart shows the time-mean**
**frequency of extreme El Niño events (the number of events per decade) for several idealised and more realistic global warming**
**experiments (abrupt-4xCO2, 1pctCO2, SSP5-8.5 and SSP1-2.6) next to the pre-industrial Control simulation (piControl). From**
**Heede and Fedorov 2023a (cropped).**

It is projected that the eastern equatorial Pacific will warm faster than the western part of the basin, leading to an eastern
equatorial Pacific (EEP) warming pattern or El Niño-like mean conditions, associated with weaker Pacific trade winds. Most
climate model future projections exhibit this pattern (e.g. DiNezio et al. 2009; Xie et al. 2010; Heede and Fedorov 2021),
and increased ENSO variability is prevalent in models that simulate stronger nonlinear (Bjerknes) feedbacks (Cai et al
2022). A recent comprehensive study of CMIP6 models and scenarios concluded that although a common mechanism to
explain a change in ENSO activity across models is missing, its increase under warming scenarios is robust (Heede and
Fedorov, 2023a).
Furthermore, during the warm Pliocene epoch approximately 3-5 million years ago when global surface temperature were
~3°C above pre-industrial, the east-west SST gradient was indeed reduced (Wara et al. 2005; Fedorov et al. 2006, 2013,
2015; Tierney et al. 2019). This state is often referred to as 'permanent El Niño-like' conditions, which does not indicate
ENSO changes but rather a consistent mean decrease in the east-west SST gradient. While debates on this topic are ongoing,
estimates for this gradient reduction range from 1.5 to 4°C, depending on the time interval, proxy data, and the definition of
this gradient.

**4.2.2 Assessment and knowledge gaps**

Therefore, there is a general expectation of a future reduction in the Pacific's east-west SST gradient by the end of the 21st
century. Together with other contributing factors, such as the strengthening of the Madden-Julian Oscillation (MJO, the

dominant intraseasonal mode in the tropical Indo-Pacific, Arnold and Randall 2015), this reduction is expected to amplify ENSO (Heede and Fedorov 2023a). Additionally, a warmer atmosphere can hold more water vapor, which could result in stronger precipitation and heating anomalies in the atmosphere, leading to greater remote impacts of El Niño events.

Consequently, the collective evidence implies an increase of El Niño magnitude and impacts under global warming. There is, however, insufficient indication for a critical transition associated with an abrupt or irreversible regime shift towards a new, more extreme, or persistent ENSO state, such that ENSO is not considered a tipping system with medium confidence (see also Armstrong McKay et al., 2022). However, it is well connected to other Earth system components (e.g. affecting tropical monsoon rainfall), thereby possibly playing a role in tipping cascades linking different tipping elements via global teleconnections (Wunderling et al., 2023).

Notably, the projections of a future EEP warming pattern, weaker mean trade winds, and stronger El Niño events contradict the recent decadal trends in the tropical climate over the past 30 years or so. In fact, since the early 1990s, the Pacific trade winds have strengthened, and the eastern equatorial Pacific has become colder (e.g. Ma and Zhou. 2016; Seager et al. 2022; Wills et al. 2022; Heede and Fedorov 2023b). Whether these trends reflect an ocean thermostat-like response to global warming, internal variability of the system, or both, remains an open question. Similarly, the magnitude of ENSO events has been generally weaker since the 2000s compared to the 1980s and 1990s (Fig. 1B; also Capotondi et al. 2015 or Fedorov et al. 2020).

Therefore, debates on the future of the tropical Pacific and ENSO revolve around the question of when the transition to a mean EEP pattern and weaker trade winds may occur, likely leading to a stronger El Niño and more frequent extreme events. Simulations with global climate models including strongly eddying ocean components (Wieners et al. 2019; Chang et al., 2020) and the recently developing 2023-2024 El Niño are expected to help reduce persistent model tropical biases in SST, precipitation, and ocean thermocline, and to resolve some of the unresolved issues.

**4.3 Mid-latitude atmospheric dynamics**

The mid-latitude atmospheric circulation is characterised by a band of strong westerly winds, with the largest velocities at an altitude of 7-12 km. In the Northern Hemisphere, these form the so-called polar 'jet-stream'. The jet serves as a separation of cold air masses at high-latitudes in the North from temperate air masses further south. Large meanders in the jet are referred to as Planetary, or Rossby waves. In most cases, these waves move over large distances and decline over timescales of a few days. When persisting for a prolonged time over the same location (referred to as 'quasi-stationary' waves) they can lead to high-impact climate extremes, including temperature extremes or heavy precipitation. An example is the record breaking heatwave of 2021 in the North American Pacific Northwest (Philip et al., 2021).

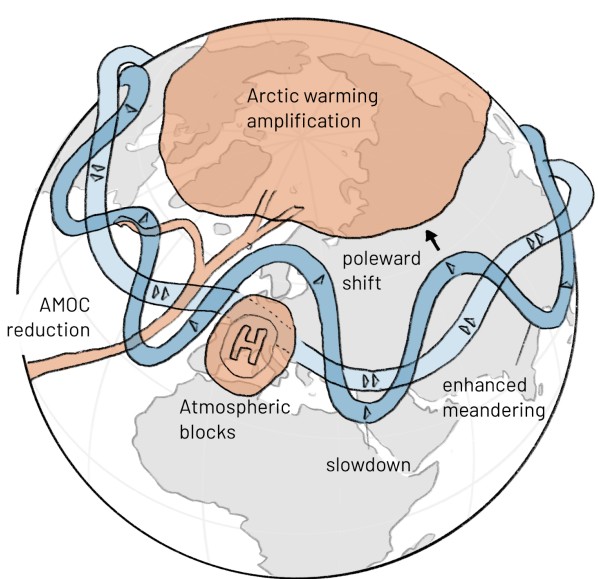

**Figure 11: Potential changes in mid-latitude atmospheric circulations under global warming, exemplary for the Northern Hemisphere. Reduction of AMOC, atmospheric blocking events, arctic warming and other drivers can modify the jet stream. Potential consequences are a northward shift, slowdown and enhanced meandering, related to increases in extreme weather phenomena.**

Atmospheric features such as blocks (quasi-stationary high-pressure regions that divert, or "block" the large-scale atmospheric flow on timescales of several days to weeks) are intimately linked to these persistent meanders in the jet. A widely discussed effect of climate change is a poleward shift of the mid-latitude jet, although this may be season and location dependent (Oudar et al. 2020), and smaller than previously thought (Curtis et al., 2020) (Figure 11).

### 4.3.1 Evidence for tipping dynamics

In climate models, the magnitude of the jet's meridional shift strongly depends on the reduction of the overturning circulation in the Atlantic (AMOC). Models with a strong AMOC reduction in the future tend to project a stronger poleward shift of the jet than models with a weaker AMOC reduction, which sometimes even display an equatorward shift(Bellomo et al., 2021). This is because a strong weakening of the AMOC might amplify the latitudinal temperature gradient that drives the strength and position of the zonal jet. As a consequence, the extent of AMOC change is a large contributor to atmospheric circulation uncertainty in regional climate change projections (Bellomo et al., 2021).

Furthermore, it has been suggested that the mid-latitude flow might weaken, leading to more persistent and slower moving weather patterns (Coumou et al. 2015, Kornhuber & Tamarin-Brodsky 2021). A possible driver is Arctic amplification, namely the fact that the Arctic is warming more rapidly than the rest of the planet, driven by a variety of mechanisms (Previdi et al., 2021) including temperature- and sea ice-related feedbacks (Dai et al., 2019). Arctic amplification reduces the equator–pole temperature contrast, and could result in a weakening and enhanced meandering of the jet stream (e.g. Francis

and Vavrus, 2015). This, in turn, could potentially lead to an even larger reduction in the latitudinal temperature contrast, further weakening the jet and so on. While Arctic amplification is most evident during winter, such increase in waviness may also be occurring during the summer season (Coumou et al., 2018). However, evidence that the occurrence of large-amplitude atmospheric waves is increasing is debated (e.g. Screen and Simmonds, 2013; Blackport and Screen, 2020; Riboldi et al., 2020), and mechanisms which would reduce blocking in the future have also been proposed (e.g. Kennedy et al., 2016).

As part of this debate, it has been proposed that several weather extremes in recent decades were associated with a quasi-stationary, quasi-resonant wave pattern. This results from the interaction of climatological waves that are perpetually forced by orography (mountain geography) and land-sea contrasts with transient meanders of the jet stream (Petoukhov et al., 2013), given a set of favourable conditions (White et al. 2022). Petoukhov et al. (2013) also hypothesised that Arctic amplification and the associated weakened, wavier jet may provide increasingly favourable conditions for the occurrence of quasi-resonance. This can result in circulation features which accelerate regional extreme weather occurrence trends, for example heatwave trends in Europe (Rousi et al. 2022), although the direction of causality is debated (Wirth and Polster, 2021). If recent extreme events are indeed associated with a resonance mechanism that only is triggered when the jet crosses a certain threshold in waviness, a tipping point might be involved. However, it is uncertain whether this would be associated with hysteresis and irreversibility or would just be a reversible, but abrupt, shift of the atmosphere towards enhanced large-amplitude mid-latitude waves.

More generally, there is no robust evidence that continued climate change and Arctic amplification will lead to a tipping towards a wavy-jet state, systematically higher amplitude and/or more frequent planetary waves, or blocks. Equally, there is no robust evidence that these hypothetical changes would be self-sustaining. Indeed, while a number of large changes in atmospheric dynamical features may occur under climate change, these are typically discussed as gradual changes, without explicit hysteresis or tipping behaviour. Similarly, there is no robust evidence pointing to tipping-like behaviour in the jet stream's latitudinal location, although gradual, long-term shifts may occur.

It should nonetheless be noted that atmospheric circulation responses to climate change are characterised by large model uncertainty and are possibly biased by the relatively low resolution of global climate models compared to e.g. weather-prediction models (Shepherd, 2019). Indeed, there is indication that for models to exhibit tipping in atmospheric blocking, they need to involve a sensitive sea-ice model and sufficient resolution to capture the sea-ice-atmosphere-AMOC feedback driving the transition (Drijfhout et al., 2013).

**4.3.2 Assessment and knowledge gaps**

Although theoretically possible, there is thus no robust evidence for tipping point behaviour in mid-latitude atmospheric circulations in the near future. At the same time, a number of relevant physical processes are currently debated or ill-constrained, not least by large model uncertainty. We thus evaluate the mid-latitude atmosphere as not displaying tipping points, with low confidence.

The mid-latitude large-scale circulation itself may nonetheless be affected by tipping behaviour of other components of the Earth system to which it is coupled. An example is the Greenland ice sheet, whose topography is important for determining the atmospheric circulation in the North Atlantic sector (White et al., 2019). A collapse of the Greenland ice sheet may trigger tipping-like changes in the atmospheric circulation in the region. The mid-latitude large-scale circulation may also contribute to tipping behaviour elsewhere in the Earth System. For example, joint non-tipping changes in mid-latitude atmospheric dynamics, the associated surface climate, and other components of the Earth system, may lead to tipping point behaviour in vegetation (Lloret and Batllori, 2021). Similarly, drier soils, a strengthened land-atmosphere coupling, and a contribution from large-scale circulation anomalies recently triggered a shift to hotter and drier conditions in inner East Asia (Zhang et al., 2020). Both the vegetation and dryness changes could in turn feed back onto the atmospheric circulation.

Due to such feedbacks and interactions between the atmospheric circulation and other components of the Earth system, and due to its role in weather and climate extremes, an improved understanding of the physical processes underlying changes in mid-latitude atmospheric dynamics under recent and future climate change appears pivotal in a tipping point context. Large model uncertainty in projecting abrupt regional atmospheric circulation changes conditioned by changes in the ocean, cryosphere, or land surface would lend itself eminently for a storyline approach (Zappa and Shepherd, 2017). Tipping of the atmospheric circulation, and associated weather extremes, would then be conditioned by threshold behaviour in other, connected systems.

Finally, we argue for the need to investigate whether recent, record-breaking weather extremes can be explained by the gradual changes in likelihood distribution due to the recent and ongoing climate change, or whether they are signs of abruptly changing likelihood distributions. Such a shift in the distribution of extremes could be diagnosed using extreme value theory. Although a shift cannot be associated with a global tipping point, it would suggest that the extreme value distribution of (a) certain type(s) of extreme weather did witness regional tipping, whether or not reversible, in the sense of a large nonlinear change in response to a small and gradual change in forcing, potentially driven by self-sustaining feedbacks.

# 5. Discussion and Outlook

In general, the circulation systems discussed in this review are driven by gradients in, e.g., temperature, salinity or pressure, and interconnect various parts of the Earth system. They are therefore very susceptible to the multitude of changes that humans are causing to all planetary spheres, disrupting established equilibria and dynamics. Alongside the significant input of additional energy to the Earth system via greenhouse gas emissions, large-scale land-use change, injection of aerosols to the atmosphere and other drivers directly and indirectly impact the atmospheric and oceanic circulation systems, all of which show signs of change (IPCC AR6, 2021).

In this review, we have investigated the potential for tipping in these systems, anchored to a tipping definition of strong positive, reinforcing feedbacks that lead to a large change beyond a forcing threshold (following Armstrong McKay et al. , 2022). Synthesis Table 1 summarises our findings, listing key drivers of change and destabilisation for each circulation

system, as well as ensuing impacts following a state shift. Importantly, we highlight a number of positive feedbacks that
could give rise to self-reinforcing change beyond a forcing threshold. Finally, we collate evidence for signature tipping
characteristics like hysteresis and abruptness, and classify each potential tipping system along the criteria defined in the
introduction.

We classify the oceanic overturning circulations - the subpolar gyre (SPG), the Atlantic Meridional Overturning Circulation
(AMOC) and Antarctic Overturning Circulation - as tipping systems with medium confidence. There are signs of a
weakening overturning in the AMOC and of the SPG circulation, and a projected decline of all three under a warming
climate. One common main driver of this change is the freshwater influx, primarily from melting of the ice sheets,
disrupting the buoyancy conditions at the overturning sites. Other contributing factors include warming oceans, and changes
in wind and precipitation patterns. While process understanding, modelling and paleo evidence suggest a possible bistability
in these systems, the assessment of exact tipping thresholds is difficult. Current climate models suffer from imperfect
representation of some important processes (such as eddies and mixing) and from biases which can impact the response to
forcings. In particular, the representation of deep convection in the Southern Ocean is deficient, with convection occurring
in the open ocean rather than shelf seas (Heuzé et al., 2021; Purich and England, 2021). It has further been shown that the
model representation of SPG tipping can be dependent on biases in stratification in the North Atlantic (Sgubin et al., 2017).
For the AMOC, various studies have suggested that it tends to be too stable in many climate models because of biases in
simulating salinity dynamics and the intertropical convergence zone (ITCZ) (Liu et al., 2017; Mecking et al 2017,
Swingedouw et al. 2022). As observational time scales are short compared to the natural variability timescales of the oceanic
overturning circulations, paleo reconstructions have proven indispensable for understanding potential long-term changes.
Indeed, the early warning methodology, increasingly applied to the North Atlantic (Boers, 2021; Ditlevsen and Ditlevsen,
2023; Michel et al., 2022), is highly dependent on these historical proxies. Albeit being subject to ongoing debate (Michel et
al., 2023), these warnings add to the reasons for concern regarding the possible tipping of ocean circulation systems.

Moving to the monsoon systems, our review highlights that there are multiple drivers beyond (direct or indirect)
anthropogenic climate change that influence these systems. Land-use change on large scales and aerosol emissions have
direct impacts on the monsoon dynamics, potentially through positive feedback loops. For example, the vegetation-albedo
coupling is identified as a critical feedback having led to past abrupt shifts in the West African monsoon (WAM), linked to
drastic changes in the Sahara vegetation states. However, limitations in modelling vegetation shifts and model disagreements
on future trends call for improved assessments. We thus consider the WAM as a tipping system, yet with low confidence.
Evidence for tipping is limited for the other monsoon systems under consideration over Asia and South America. Hence, we
attribute an uncertain status to them. Studies on the Indian summer monsoon (ISM) tipping so far have employed simplified
analytical models that fail to represent the full spectrum of feedbacks and processes within this complex monsoon system
(Boos and Storelvmo, 2016; Kumar and Seshadri, 2023). Similar arguments apply to the East Asian summer monsoon
(EASM). Current climate models further inadequately capture South American monsoon (SAM) precipitation, and the

impact of climate change on SAM precipitation remains uncertain (Jones and Carvalho, 2013; Douville et al., 2021). Further research utilising paleoclimate evidence, climate model reconstructions of past monsoons, and comprehensive modelling of the system's responses to various forcing thresholds are required to clarify any non-linear changes within both the Indian and South American monsoon systems. Critically, the dynamics of the global monsoon are substantially influenced by the AMOC, the position of the ITCZ and interhemispheric aerosol loadings, such that a global, coupled approach is promising for more insights into potential tipping of the monsoon systems.

Finally, we have considered global atmospheric circulation systems. Although theoretically possible, there is no robust evidence for tipping point behaviour in mid-latitude atmospheric circulations. Nonetheless, there are some physical mechanisms, such as enhanced jet stream meandering and planetary wave resonance, which may lead to tipping-like behaviour. However, their physical interpretation and relevance for tipping are debated. In more general terms, the atmospheric circulation responses to climate change are characterised by large model uncertainty (Shepherd, 2019), and mid-latitude circulations may be affected by tipping behaviour of other components of the Earth system (Orihuela-Pinto et al., 2022). We thus categorise the mid-latitude atmosphere as not displaying tipping points, albeit with low confidence. Similarly, there is insufficient indication for a state shift in El Niño Southern Oscillation (ENSO) towards a state with more persistent or extreme El Niño events. An increase of El Niño magnitude and impacts is expected under global warming, however, not abruptly or irreversibly. For these reasons and a lack of a comprehensive mechanism across models to explain a change in future ENSO activity (Heede and Fedorov, 2023a), we classify ENSO as no tipping system with medium confidence. Residual uncertainties remain due to divergence of model predictions, and known model biases in SST, precipitation and ocean thermocline (Wieners et al. 2019; Chang et al., 2020). With similar confidence we can rule out cloud-driven tipping points. Although large scale reorganisations are in principle conceivable, conjectures of strong positive feedbacks leading to extreme climate sensitivity, e.g. runaway warming or rapid disappearance of low clouds, are deemed unlikely based on current models and understanding (LeConte et al. 2013). In summary, we rate the concern about tipping of these global circulation systems as low with medium to low confidence. This, however, does not rule out non-linear amplification of trends or interactions with other tipping systems in the Earth system, like the high-latitude cryosphere.

The review presented here was conducted by an expert group, drawing on multiple lines of evidence, including observational data, coupled models of low and high complexity and paleo records. It presents a broad overview of the present state of knowledge on the subject of circulation tipping points, while acknowledging the challenges and limitations discussed below

One general difficulty in assessing tipping points is the inherent need to map out and quantify relevant amplifying and dampening feedback loops. Naturally, this requires understanding and modelling nonlinearities of the underlying systems, which becomes increasingly challenging conceptually when the relevant feedbacks involve many diverse systems (as they typically do for coupled ocean and atmosphere circulations). This conceptual issue translates into a numerical one, where model complexity needs to be traded-off against computational runtimes. Partly for these reasons, there is currently only a

limited evidence base on systematic assessments of destabilising drivers on potential tipping systems, such as NAHosMIP exploring freshwater impacts on the AMOC (Jackson et al., 2023).

In recent years, however, there has been a surge of numerical experiments with models of ever-increasing complexity and resolution (e.g., Liu et al., 2017, Mecking et al 2017, Chang et al., 2020, Li et al., 2023). These experiments shed light on so far under-represented important feedback processes and small-scale dynamics, and identify potential model biases towards simulating too stable conditions. Model resolution will affect the type of instabilities which can be represented. For example, on ocean circulations, once the spatial resolution of an ocean model is smaller than the internal Rossby deformation radius, baroclinic instabilities can be represented leading to ocean eddies. The presence of these eddies will also lead to modification of the mean flow and associated transports, and hence affect its stability properties. However, this has only been systematically analysed for very idealised cases, such as wind-driven ocean circulation (Berloff and McWilliams, 1999). Moreover, there was no relationship found between the presence of weak AMOC states (that are stable for at least a hundred years) and ocean resolution in a set of CMIP6 GCMs (Jackson et al 2023). Therefore, although higher resolution captures important dynamics, its effect on model stability remains an active area of investigation. Model complexity, if meant as to include additional processes, can strongly affect the stability of an ocean flow. For example, by including a sea-ice component into an ocean model, the multi-stability regime of the AMOC will be affected (Van Westen and Dijkstra, 2023; Van Westen et al., 2024). Along similar lines, recent work demonstrates that high-resolution modelling of ocean eddies impacts mid-latitude atmospheric patterns and storm tracks (Zhou and Cheng, 2022; Czaja et al., 2019), albeit with no explicit investigation on potential tipping.

Building on this progress, with respect to tipping in ocean and atmosphere circulations, future research should intensify its efforts to address these key research questions:

(1) Are circulation systems susceptible to tipping? Where tipping is in principle conceivable, is its absence in numerical simulations a deficit of the model or can physical stabilising feedbacks be identified? This involves bridging the gap between physical process understanding and conceptual models (which clearly allow for tipping) on one hand, and high-complexity and -resolution numerical models that might be biased towards stability on the other hand. Paleo reconstructions provide an essential resource for this, along with continued observations at high spatial and temporal resolutions.

(2) At which levels of forcing are tipping thresholds expected based on systematic uncertainty assessment? Such an analysis should take into account different forms of uncertainties such as on model initial conditions, parameterisation and model structure as well as in data products from Earth observations and paleoclimate reconstructions. New methodologies need to be developed to propagate these uncertainties transparently towards providing reliable and interpretable estimates of tipping threshold distributions.

(3) What are the impacts of tipping point transgressions? Tipping of any of the discussed systems would entail significant consequences for Earth and human systems. As for the AMOC, a state shift can cause disruptions in

many other systems across the globe, including shifts in precipitation patterns in the Northern hemisphere and in
the tropics with significant expected impacts on agriculture, human settlements and other infrastructure adapted to
pre-tipping climate conditions. An understanding of the tipping dynamics and the anticipated biophysical states
after tipping can serve as foundation for standardised impact modelling, or for storyline approaches on extreme
climate change.

An initiative designed to explicitly investigate these research questions in a systematic way is the now established Tipping
Points Modelling Intercomparison Project (TIPMIP; Winkelmann et al., 2025). To that end, it is designed to study the
occurrence of tipping across a range of biophysical systems, investigating the critical thresholds and aspects like
irreversibility, rate-induced tipping and spatio-temporal scales. TIPMIP builds on domain-specific endeavours (e.g.
NAHosMIP, ISMIP, …), thereby complementing general modelling initiatives addressing parts of these questions in ongoing
programs (e.g. CMIP). In addition, updated expert elicitations on e.g. tipping interactions (Kriegler et al., 2008) can serve as
a valuable complement to numerical modelling efforts. Overall, the multi-faceted questions on the tipping potential of
circulation systems may be answered by studying the hierarchy of models. This way, the effects of small-scale processes
(model resolution) and additional large-scale feedback processes  (model complexity) on the tipping behaviour can be
systematically assessed (Dijkstra, 2013). Relatedly, theoretical advances shed light on new properties to be studied in tipping
systems such as a shifted focus from stable long-term attractors to the change in relative basin widths when studying
rate-induced tipping (Ashwin et al., 2012; Feudel, 2023). The former is anyways often-times regarded as oversimplification
that does not fully hold up to time-varying, continuous drivers (Bathiany et al., 2018) or in systems with spatial degrees of
freedom (Rietkerk et al., 2004). Steps forward can build on insights from hydrodynamic theory such as critical transitions in
turbulence on multiple scales (van Kan and Alexakis, 2020; van Kan, 2024) or transitions between multiple- stable states in
aquaplanets (Brunetti et al., 2019; Ragon et al., 2022). Of particular importance in the context of circulation systems is the
role of interactions between different parts of the Earth system, and the implications for reciprocal (in-)stability and aspects
like early warning signals (Klose et al., 2021). Overall, it is needed to advance not only resolution and complexity of
numerical models but also to further pursue fundamental, conceptual research in nonlinear Earth system dynamics for a
better understanding of the underlying mechanisms and relevant uncertainties in climate tipping points. Finally, an improved
understanding of how to monitor tipping systems is needed, to eventually establish early warning systems for mitigation and
adaptation purposes (Swingedouw et al., 2020; Wood et al., 2024).

Table 1. Summary of evidence for tipping dynamics, key drivers, and biophysical impacts in each system considered in this
review. Key: +++ Yes (high confidence), ++ Yes (medium confidence), + Yes (low confidence), – – – No (high confidence), –
– No (medium confidence), – No (low confidence)  Primary drivers are bolded, DC: Direct Climate driver (via direct
impact of emissions on radiative forcing); CA: Climate–Associated driver (including second–order & related effects of
climate change); NC: Non–Climate driver, PF: positive (amplifying) feedback, NF: negative (damping) feedback. Drivers
can enhance (↑) the tipping process or counter it (↓)

| Key drivers | Key biophysical impacts | Key feedbacks | Abrupt / large rate change? | Critical threshold(s)? | Irreversible? (timescale) | Tippin system |
|---|---|---|---|---|---|---|
| **OCEAN CIRCULATIONS** | | | | | | |
| **Atlantic Meridional Overturning Circulation (AMOC)** Shutdown/collapse | | | | | | |
| • **DC: ocean warming (↑)**<br>• **DC: precipitation increase (↑)**<br>• **CA: Greenland ice sheet meltwater increase (↑, primary in the future)**<br>• **CA: Arctic river discharge increase (↑)**<br>• CA: sea ice extent & thickness decrease (↑) | • Cooling over Northern Hemisphere (up to 10°C over W/N Europe)<br>• Warming of the surface South Atlantic<br>• Change in precipitation and weather patterns over Europe<br>• Change in location and strength of rainfall in all tropical regions<br>• Reduced efficiency of global carbon sink, and ocean acidification<br>• Reduced support for primary production in Atlantic oceans<br>• Deoxygenation in the North Atlantic<br>• Change in sea level in the North Atlantic<br>• Modification of sea ice and arctic permafrost distribution<br>• Increase in winter storminess in western Europe<br>• Southward migration of the Atlantic subtropical gyres<br>• Reduced land productivity in western Europe<br>• Increased wetland in some tropical areas and associated methane emission<br>• Change in rainforest response in drying regions | • **Salt–advection (↑)**<br>• Sea-ice melting (↑)<br>• Heat transport (↓)<br>• Temperature (↑)<br>• Surface heat flux (↑)<br>• Collapse of convection in the Labrador and Irminger Seas (↑) | Feedback-dependent:<br><br>Century (basin-wide salt advection feedback),<br><br>Few decades (North Atlantic salt-advection feedback),<br><br>< few decades (sudden increase in sea-ice cover in all convective regions) | **Salinity change/freshwater/AMOC strength**<br><br>Thresholds likely path-dependent (depending on rate and spatial pattern) | ++ (centuries) | ++ |
| **North Atlantic Subpolar Gyre (SPG)** Collapse | | | | | | |

| Key drivers | Key biophysical impacts | Key feedbacks | Abrupt / large rate change? | Critical threshold(s)? | Irreversible? (timescale) | Tippin system |
|---|---|---|---|---|---|---|
| • **DC: ocean warming (↑)**<br>• **DC: precipitation increase (↑)**<br>• **CA: Greenland ice sheet meltwater increase (↑, primary in the future)**<br>• **CA: Arctic river discharge increase (↑)**<br>• CA: sea ice extent & thickness decrease (↑) | • Increase in summer heat waves frequency<br>• Collapse of the North Atlantic spring bloom and the Atlantic marine primary productivity<br>• Increase in regional ocean acidification<br>• Regional long-term oxygen decline<br>• Impact on marine ecosystems in the tropics and subtropics | • **Salt–advection (↑)**<br>• Sea-ice melting (↑)<br>• Heat transport (↓)<br>• Temperature (↑)<br>• Surface heat flux (↑)<br>• Collapse of convection in the Labrador and Irminger Seas (↑) | Years to few decades | **Salinity change/freshwater**<br><br>Global warming 1.1-3.8°C | ++ (decades) | ++ |
| **Southern Ocean circulation** Antarctic Overturning Collapse / Rapid continental shelf warming | | | | | | |
| • **DC: ocean warming (↑)**<br>• **CA: Antarctic ice sheet meltwater increase (↑)**<br>• CA: wind trends (↑)<br>• CA: Sea ice formation (↑)<br>• DC: precipitation increase (↑) | • Modification of Earth's global energy balance, timing of reaching 2°C global warming<br>• Reduced efficiency of global carbon sink<br>• Change in global heat storage<br>• Reduced support for primary production in world's oceans<br>• Drying of Southern Hemisphere<br>• Wetting of Northern Hemisphere<br>• Modification of regional albedo, shelf water temperatures<br>• Potential feedbacks to further ice shelf melt | • Density–stratification (↑)<br>• Meltwater-warming (↑) | ++ (AABW formation & abyssal overturning shutdown within decades) | Salinity change/freshwater | ++<br><br>(cavity warming reversion would need 20th-century atm conditions + reduced meltwater input) | ++ |

<br>

| ATMOSPHERIC CIRCULATIONS: MONSOONS |
|---|

| **Indian summer monsoon (ISM)** Collapse / Shift to low-precipitation state |
|---|

| Key drivers | Key biophysical impacts | Key feedbacks | Abrupt / large rate change? | Critical threshold(s)? | Irreversible? (timescale) | Tipping system |
|---|---|---|---|---|---|---|
| <ul><li>NA: increased summer insolation (↓)</li><li>DC: increased water vapour in atmosphere (↓)</li><li>CA: Indian Ocean Dipole events (?)</li><li>CA: ENSO change (?)</li><li>CA: North Atlantic cold SST (↑)</li><li>NC: aerosol loading (↑)</li><li>CA:Indian Ocean warming (↑)</li><li>CA: low cloud reduction (↓)</li></ul> | <ul><li>**Massive change in precipitation**</li><li>Change in tropical and subtropical climates</li><li>Biodiversity loss and ecosystem degradation</li></ul> | <ul><li>Moisture-adve ction (↓)</li></ul> | Decades to centuries | Regional AOD level over Indian subcontinent (>0.25)<br><br>Interhemispheric AOD difference (>0.15)<br><br>AMOC slowdown (unknown threshold) | Uncertain; likely decades to centuries | **unknown** |
| **West African monsoon (WAM)** Collapse or abrupt strengthening | | | | | | |
| <ul><li>DC: increased water vapour in atmosphere</li><li>NA: increased summer insolation</li><li>NC: land-cover change</li><li>CA: desertification</li><li>CA: AMOC slowdown</li><li>CA: regional SST variations</li><li>CA: High latitude cooling</li><li>CA/NC: regional soil moisture variation</li><li>CA/NC: regional vegetation variation</li><li>NA: dust emissions</li></ul> | <ul><li>**Massive change in precipitation**</li><li>Change in tropical and subtropical climates</li><li>Biodiversity loss and ecosystem degradation</li></ul> | <ul><li>Vegetation-al bedo (↑)</li></ul> | Decades to centuries | Insolation changes in the Northern Hemisphere summers and surface albedo changes (unknown threshold)<br><br>Interhemispheric asymmetry in AOD (>0.15)<br><br>AMOC slowdown (unknown threshold) | Decades to centuries | + |
| **South American monsoon (SAM) Collapse or abrupt strengthening** | | | | | | |
| <ul><li>DC: increased water vapour in atmosphere</li><li>NA: increased summer insolation</li><li>CA: AMOC slowdown</li><li>NC: Amazon deforestation</li></ul> | <ul><li>**Massive change in precipitation**</li><li>Change in tropical and subtropical climates</li><li>Biodiversity loss and ecosystem degradation</li></ul> | <ul><li>Vegetation-moi sture (?)</li></ul> | Decades | Interhemispheric asymmetry in AOD (>0.15)<br><br>Extent of Amazon deforestation (30-50%)<br><br>AMOC slowdown (unknown threshold) | Uncertain; likely decades to centuries | **unknown** |

| |
|---|
| **GLOBAL ATMOSPHERIC CIRCULATIONS** |
| **Tropical clouds, circulation and climate sensitivity** Shift to different large-scale configuration |

| | Key drivers | Key biophysical impacts | Key feedbacks | Abrupt / large rate change? | Critical threshold(s)? | Irreversible? (timescale) | Tippin system |
|---|---|---|---|---|---|---|---|
| | • **DC: atmospheric warming (↑)**<br>• **DC: ocean warming (↑)** | • Massive alteration of hydrology in many regions<br>• Impact on ambient atmospheric-oceanic phenomena such as ENSO<br>• Strong intensification of global climate change | • Cloud-moisture-radiation (↑) | Unknown | Unknown | Unknown | -- |
| **El Niño Southern Oscillation (ENSO)** Shift to more extreme or persistent state | | | | | | | |
| | • **DC: east vs west Pacific warming (↑)**<br>• **DC: increased water vapour in atmosphere (↑)**<br>• DC: weaker trade winds (↑)<br>• CA: MJO strengthening (↑) | • Temporary trade wind collapse during El Niño phase<br>• Increase in global mean surface temperatures during El Niño phase<br>• Modification of global atmospheric circulation<br>• Modification of worldwide patterns of weather variability | • Bjerknes (↑) (SST-tradewinds-ocean thermocline) | No evidence (gradual) | No evidence (gradual) | No evidence | -- |
| **Mid-latitude atmospheric dynamics** Shift to wavy-jet state / more frequent or extreme planetary waves or blocks / latitude shift | | | | | | | |
| | • CA: AMOC slowdown<br>• CA: Midlatitude flow weakening (↑)<br>• DC: Arctic amplification (↑) | • More persistent and slower moving weather patterns<br>• Increase in extreme events on Northern hemisphere | • Debated: Waviness quasi-resonance (↑) | No evidence | Potentially waviness threshold, beyond which quasi-resonance kicks in | No evidence | - |

*Code and data availability.* There is no data or code that has been produced for this review article.

*Author contributions.* SL and D.I.A.M have designed the overall structure and scope. All authors have contributed to the writing. SL and YA produced the figures with input from the authors. All authors have reviewed and edited the final version of the manuscript.

*Competing Interests.* Some authors are members of the editorial board of the journal Earth System Dynamics. Otherwise, the authors have no other competing interests to declare.

*Acknowledgements*. This review article has been carried out within the framework of the Global Tipping Points Report 2023. We would like to thank Norman Steinert for developing the original concept of the tipping systems overview figure (Fig 1) and for providing the basemap. S.L. and D.I.A.M. acknowledge financial support from the Earth Commission. The work of the Earth Commission, a program of Future Earth, is made possible through the support of the Global Commons Alliance, a sponsored project of Rockefeller Philanthropy Advisors. Furthermore, D.I.A.M. acknowledges financial support via the Global Tipping Points Report project, funded by the Bezos Earth Fund. Y.A., S.R. and B.S. acknowledge funding from the project COMFORT (grant agreement no. 820989) under the European Union's Horizon 2020 research and innovation programme, and from the EC Horizon Europe project OptimESM "Optimal High Resolution Earth System Models for Exploring Future Climate Changes", grant 101081193 and UKRI grant 10039429, CANARI (NE/W004984/1), UKRI/NERC HighLight Topic Projects "Interacting ice Sheet and Ocean Tipping - Indicators, Processes, Impacts and Challenges (ISOTIPIC)", grant NE/Y503320/1, and "Consequences of Arctic Warming for European Climate and Extreme Weather" (ArctiCONNECT, NE/V004875/1). Y.A. and S.R. also acknowledge funding support from the UK NERC projects LTS-M BIOPOLE (NE/W004933/1). The work of Y.A. is also funded by the UKRI (grant 10038003) as part of the EPOC project (Explaining and Predicting the Ocean Conveyor; grant number: 101059547). Views and opinions expressed are however those of the author(s) only and do not necessarily reflect those of the European Union. Neither the European Union nor the granting authority can be held responsible for them. Y.A. and S.R. acknowledge the use of the ARCHER UK National Supercomputing and JASMIN. For the use of the data from the NEMO-MEDUSA 1/4 degree high resolution model simulations in this paper we acknowledge authorship of Drs Andrew Coward, Andrew Yool, Katya Popova and Stephen Kelly from the National Oceanography Centre UK (NOC). D.S. acknowledges funding from the project TipESM, EU grant 10111337673 and Advanced Research + Invention Agency (ARIA)'s UK projects SORTED and AdvanTIP (SCOP-PR01-P003 - Advancing Tipping Point Early Warning AdvanTip). For the EU projects the work reflects only the authors' view; the European Commission and their executive agency are not responsible for any use that may be made of the information the work contains. C.M.C. acknowledges the financial support from FAPESP (grants 2018/15123-4 and 2019/24349-9) and CNPq (grant 305285/2025-4). HD acknowledges financial support from the European Research Council through the ERC-AdG project TAOC (project 101055096). J.F.D. acknowledges financial support from the European Research Council Advanced Grant project ERA (Earth Resilience in the Anthropocene, ERC-2016-ADG-743080) and by the project CHANGES funded by the German Federal Ministry for Education and Research (BMBF) within the framework 'PIK_Change' under grant 01LS2001A. M.H.E. acknowledges support from the Australian Research Council (ARC), Grant Nos. SR200100008 and DP190100494. A.V.F. acknowledges support from the ARCHANGE project of the "Make our planet great again" program (ANR-18-MPGA-0001), France, and from National Aeronautics and Space Administration (80NSSC21K0558), National Oceanic and Atmospheric Administration (NA20OAR4310377), National Science Foundation (AGS-205309), and Department of Energy (DE-SC0023134, DE-SC0024186), USA. LJ was supported by the Met Office Hadley Centre Climate Programme funded by DSIT and by TiPES. This project is TiPES contribution no. 262, and this project has received funding from the European Union's Horizon 2020 research and innovation programme under grant

agreement no. 820970. GM acknowledges funding from the Swedish Research Council Vetenskapsrådet (proj. No. 2022-06599). T.T acknowledges funding from the Department of Science and Technology, India through the 'INSPIRE Faculty Fellowship'. This is ClimTip contribution 40, the ClimTip project has received funding from the European Union's Horizon Europe research and innovation programme under grant agreement No. 101137601: Funded by the European Union. Views and opinions expressed are however those of the author(s) only and do not necessarily reflect those of the European Union or the European Climate, Infrastructure and Environment Executive Agency (CINEA). Neither the European Union nor the granting authority can be held responsible for them. JBS was supported by OCEAN ICE, which is funded by the European Union, Horizon Europe Funding Programme for research and innovation under grant agreement Nr. 101060452.

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
