# Peer review of "Tipping points in ocean and atmosphere circulations"

_EGUsphere, 2023_

## Author Response (AR1)

**Author's response to *"Tipping points in ocean and atmosphere circulations"**

We thank Axel Kleidon for handling our manuscript and the reviewers for their feedback to improve our manuscript, which we have revised accordingly. Below are the original comments from reviewers in black and our corresponding responses in blue.

Sina Loriani for the author team

**Referee #1**

Thank you for investing the time and effort to review this manuscript - especially given its length and range of discussed systems - and for your thorough and critical assessment.

**The review "Tipping points in ocean and atmosphere circulations" discusses the current state of research regarding potential "tipping points" in the Earth system that may be inherent in circulation systems. The topic is certainly important and suitable for the journal. The aim and content of the study is clear, but could be better justified; and could also become a bit more ambitious. My main concern in this context is that there has been a large number of reviews about Earth system "tipping points" in recent years, e.g. Lenton et al., 2008 (PNAS), Lenton 2013 (Annual Review of Environment and Resources), Schellnhuber et al., 2016 (Nat Clim Change), Bathiany et al., 2016 (Dyn Stat Clim Sys), Steffen et al., 2018 (PNAS); Boers et al., 2022 (ERL), Wang et al., 2023 (Rev Geophys), Armstrong McKay et al., 2022 (Science), and a several 100 pages long Global Tipping Points Report (2023). It sometimes almost seems like scientists would spend more time on such reviews than they spend on actually creating new knowledge and reducing the uncertainties. To offer new insights, a new review paper should find its own angle and focus on content that goes beyond previous reviews. I think that the focus on circulation-related tipping elements is a promising idea, in particular regarding the atmospheric circulation. I believe that there is a bit more to discuss than mainly repeating contents from previous studies.**

Indeed, the body of literature on the topic of tipping points is steadily growing across different Earth system domains and disciplines. The strength of this author team lies in its specific expertise on the different circulation systems both in the atmosphere as well as the ocean, hence providing a unique perspective on potential tipping in these systems. The team has originally assembled to write the corresponding chapter in the Global Tipping Points Report (2023) mentioned above, and submitted this here to be peer reviewed for further scrutiny. By assessing all the different circulation systems with one approach based on identifying self-reinforcing feedbacks beyond a critical threshold, we believe we can provide a valuable contribution in sorting the wide literature on this topic. We acknowledge that the justification in the introduction was too sparse, which we expanded on now in the resubmitted manuscript.

**Potentially interesting questions the authors could discuss in more detail are: Are circulation systems more or less prone to "tipping" than terrestrial systems or ice sheets, and why?**

Arguably, the timescales on which potential tipping in the circulation system operates tend to be shorter when compared to ice sheets. However, we are not aware of a systematically larger sensitivity to tipping. On the contrary, where available, estimates of critical thresholds are generally lower for large parts of the ice sheets than, say, AMOC. The relatively low abundance of literature on tipping in mid/high-latitude atmospheric circulations compared to other systems is also not indicative of a high propensity to tipping. However, it may also be indicative of research gaps, particularly when considering more complex dynamical transitions (e.g. transitions between different chaotic attractors) possible in atmosphere-ocean variability that may be triggered by anthropogenic forcing.

**There are many insights from hydrodynamical theory about nonlinear regime shifts, from the very conceptual Lorenz system and fundamental phenomena like the onset of turbulence in fluids, to hydrodynamic instabilities of flows (barotropic and baroclinic instabilities), and the phenomena of abrupt monsoon onsets (as part of the annual cycle) on aquaplanets and the present-day Earth. Can these insights inform us on potential tipping points in atmospheric circulation under greenhouse (and/or other anthropogenic) forcing?**

Due to the complexity of the problem, tipping point science needs to rely on a multitude of evidence lines, ranging from observational and paleo(-proxy) indications to models across the hierarchy. In particular, fundamental insights about potential tipping stem from theoretical exercises and in turn, theoretical studies of tipping and regime transitions even refer to oceanic circulations as prime examples (e.g. Feudel, 2023, Bathiany et al., 2018). We agree that more theoretical insights from hydrodynamic theory such as critical transitions in turbulence on multiple scales (van Kan and Alexakis, 2020; van Kan, 2024) or transitions between multiple- stable states in aquaplanets (Brunetti et al., 2019; Ragon et al., 2022) can inform a better understanding of more applied research on climate tipping points going forward. We will add a corresponding discussion and literature references to the revised manuscript.

**Why do the authors cover the major monsoon systems but not the East Asian summer monsoon? There is evidence about abrupt shifts in this monsoon system in paleo records, e.g. Wang et al., 2008.**

In this paper, we originally only discussed the tropical regional monsoons. The East Asian "monsoon" (EASM) is mostly extratropical. Unlike classic monsoons driven by the ITCZ and characterised by heavy summer rainfall, the precipitation and wind patterns of this "monsoon" are linked to mid-latitude frontal systems and the jet stream, and its interaction with the Tibetan Plateau (Molnar et al., 2010; Son et al., 2019). The dynamics of the East Asian "monsoon" thus differs from the typical monsoons (Molnar et al., 2010; Son et al., 2019). We have now added some context in the monsoon summary section 3.4. As suggested by the reviewer, we have summarised the potential tipping behaviour of the EASM as evidenced from paleoclimate records and analyses based on the proxy record.

**Can we transfer knowledge from one monsoon system to another, or are they too different?**

Traditionally viewed as giant sea breeze circulations linked to land-sea temperature differences, monsoons are now understood as part of a global monsoon system that significantly influences the annual patterns of precipitation in these regions, driven by seasonal shifts in atmospheric circulation and the movement of convergence zones, with local variations arising from specific surface conditions (Geen et al., 2020). Hence, monsoons are complex phenomena resulting from the interplay of local and global factors, with the dominant control lying in the large-scale dynamics of the tropical atmospheric circulation and its seasonal migration. Regional attributes such as geography, topography, and land-sea distribution significantly influence the specific characteristics and behaviour of each monsoon, and their responses to climate change. For example, Ben-Yami et al. (2024) find consistent patterns in how monsoons respond to an AMOC collapse, and the Indian and West African Monsoons show an overall decrease in rainfall, whereas the impact on South American monsoon is spatially dependent (i.e., increase in rainfall in the southern domain and decreased rainfall in the northern domain of the South American monsoon). We suggest that while knowledge can be shared between monsoon systems, the effects of their unique regional traits must be carefully considered.

**What is the (nonlinear) response of the atmospheric circulation to topography? What happens when ice sheets shrink? Do we have evidence for that from reconstructions (e.g. dust record in Greenland ice core)?**
Thank you for pointing this out. We had originally indicated a link between high-latitude cryosphere and the state of mid/high-latitude atmospheric circulation, which we elaborate more on in the resubmitted manuscript.

**How does model resolution and complexity affect the stability of circulation systems? Are there any hypotheses about that, maybe not in general, but regarding ocean eddies?**
Model resolution will affect the type of instabilities which can be represented. For example, on ocean circulations, once the spatial resolution of an ocean model is smaller than the internal Rossby deformation radius, baroclinic instabilities can be represented leading to ocean eddies. The presence of these eddies will also lead to modification of the mean flow and associated transports, and hence affect its stability properties. However, this has only been systematically analysed for very idealised cases, such as wind-driven ocean circulation (Berloff and McWilliams, 1999). However there was no relationship found between the presence of weak AMOC states (that are stable for at least 100 years) and ocean resolution in a set of CMIP6 GCMs (Jackson et al 2023). Model complexity, if meant as to include additional processes, can strongly affect the stability of an ocean flow. For example, by including a sea-ice component into an ocean model, the multi-stability regime of the AMOC will be affected (Van Westen and Dijkstra, 2023; Van Westen et al., 2024). Along similar lines, recent work demonstrates that high-resolution modelling of ocean eddies impacts mid-latitude atmospheric patterns and storm tracks (Zhou and Cheng, 2022; Czaja et al., 2019), albeit with no explicit investigation on potential tipping.

**Clouds seem to show nonlinear behaviour in several ways; how precisely would this translate to climate timescales and tipping behaviour? Isn't the high spatio-temporal variability of cloud formation and dissolvement already an argument against tipping points, at least against multiple alternative states?**
We agree with this assessment. We considered it worthwhile though to include a short

discussion of clouds, since the possibilities of large-scale reorganisation are sometimes brought up, meriting a short classification from the tipping points perspective.

**What needs to happen precisely to answer these questions (beyond general statements like improving models and data).**

One way is to systematically study the behaviour of the tipping behaviour in a hierarchy of models of increasing complexity. In this way, the effects of small-scale processes (model resolution) and additional large-scale feedback processes (model complexity) on the tipping behaviour can be systematically assessed. A first sketch of this approach can be found in chapter 6 of the book Nonlinear Climate Dynamics (Dijkstra, 2013) to which we refer now in the revised paper. Systematic efforts like TIPMIP (building on initiatives like NaHosMIP, ISMIP,...) provide the evidence base to include high-complexity models in such approaches. Relatedly, theoretical advances shed light on new properties to be studied in tipping systems such as a shifted focus from stable long-term attractors to the change in relative basin widths when studying rate-induced tipping (Aswhin et al., 2012; Feudel, 2023). The former is anyways often-times regarded as oversimplification that doesn't fully hold up to time-varying, continuous drivers (Bathiany et al., 2018) or in systems with spatial degrees of freedom (Rietkerk et al., 2004). Of particular importance in the context of circulation systems is the role of interactions between different parts of the Earth system, and the implications for reciprocal (in-)stability and aspects like early warning signals (Klose et al., 2021). Overall, it is needed to advance not only resolution and complexity of numerical models but also to further pursue fundamental, conceptual research in nonlinear Earth system dynamics for a better understanding of the underlying mechanisms and relevant uncertainties in climate tipping points.

**In my view, a second major caveat of the current draft is that the labels of "tipping" potential and the uncertainty, as summarised in Fig. 1 and Table 1, involve too much subjectivity. The authors write that the assessment "… was conducted by an expert group and does not necessarily represent the view of the entire community." I believe that one can do better than that in a review that is meant to represent the general state of knowledge, and not only the author's perspective alone - either by accepting that uncertainties are not quantifiable and hence not providing any label for the plausibility of tipping, or by formulating specific criteria that would be transformed into these labels transparently.**

**Some of the confidence levels are not convincing to me given the large uncertainties of all tipping elements, in particular the "++" labels for the ocean circulation cases. For example, as a layperson, I would interpret the label "Tipping System (confidence level): yes (medium)" as the statement that it is scientifically relatively clear that these systems have tipping points. But this is not at all the case.**

Thank you. We argue that any work of this sort in the end boils down to the (subjective) assessment of the author team, but agree that the assessment and review criteria should be outlined more clearly and transparently, which we have now done in the beginning of Section 5. Indeed, the "++" label means that the science is quite indicative that the system has a tipping point ("medium to robust evidence with medium to high agreement that this system features tipping dynamics"), which in our view applies to the ocean circulations based on paleo-climatic evidence and the demonstration of tipping points across the model hierarchy.

**Another example is that the authors write "we classify the West African monsoon as a tipping system", but it is unclear to me why particularly this monsoon, and less so other monsoon systems (let alone the East Asian monsoon, which is largely ignored although it also shows nonlinear shifts in paleo records)?**

We now added some more summarising explanations in the text. For WAM, it is the combination of paleo records as well as the identification of strong, reinforcing feedbacks (see our definition of tipping points now added to the introduction) that makes us assess WAM as a tipping system, albeit with low confidence. We also added a short note on East Asian monsoon.

**Third example: If there are models that show "tipping" of blocking behaviour (Sect 4.3.1), why is the author's assessment "no tipping" with low confidence (Sect 4.3.2), while the AMOC is labeled as "tipping" with medium confidence?**

In this specific case, we are aware of one paper for blocking (from 2013), as opposed to a significant body of literature concerning AMOC, with lots of complementary sources (paleo reconstructions and models across the hierarchy, potentially even observations via proxies). Unless there is a significant bias in literature highlighting that tipping of blocking is severely understudied, we assume the available evidence supports these confidence statements.

**Specific minor comments**

**Abstract. "Evidence about tipping of the monsoon systems over South America and Asia is limited – however, there are multiple potential sources of destabilisation, including large-scale deforestation, air pollution, and shifts in other circulation patterns (in particular the AMOC)". Change of topic in mid-sentence, from monsoons to AMOC.**

The sentence aims to highlight the complex interplay of factors affecting monsoon stability. While the AMOC is not a direct driver of monsoons, research suggests its variability can influence large-scale atmospheric circulation patterns that, in turn, impact monsoon systems. We have now rewritten the sentence to reflect this more clearly.

**Introduction. Unclear: The focus is on tipping points in the future, due to human activity? This is implied by the text but not clearly stated anywhere in the abstract or introduction. The introduction is extremely short. I was expecting some more background and a justification why the authors focus on circulation systems (e.g. as opposed to tipping of ice sheets or ecosystems), and the rationale and structure of the article. One could also explain the methods (how was literature collected / considered, how were assessment criteria defined). What is considered "tipping" in this article? "shift to a different state" is vague, and cited papers disagree in their definition. This paper should stick to a certain definition.**

Thank you for pointing these aspects out. We initially kept the introduction fairly short since the manuscript is quite long anyways and each system is introduced individually. But we agree that a better rationale and specification of working definitions is needed, which we added now. We hope this reasoning also supports the response to your first comment: One major value in such a review article lies in the application of one classification scheme by an expert team of authors to sort the heterogeneity in the definitions in literature you mentioned.

**Fig. 1:  tipping under human forcing?**

We added the definition of a tipping system in the introduction.

**I 149-150: "nor the presence of external forcings such as increasing greenhouse gases" – The authors do take greenhouse gases into account (otherwise the AMOC would not destabilise), though indeed in a rather simplistic way (stable forcing and then linear increase with extrapolation into the future).**
Agreed, rewritten.

**I 145-155: "However, the claim that we might expect tipping within a few decades is – in the view of the present authors – not substantiated enough." I agree with the authors. However, rather than elaborating mostly on the "pro tipping" literature and then saying that one does not agree, it would be more convincing to let the literature speak for itself, and also elaborate a bit on the evidence against the tipping hypothesis. This would be more suitable for a review paper than just making a subjective statement.**
We cited the literature that comes to these conclusions, and removed the subjectivity disclaimer.

**I 164: Sentence is unclear. What does "these" refer to?**
Meant were small-scale processes, added.

**I 170: "even the current generation of climate models have quite low spatial resolution and do not characterise narrow currents, eddies and processes such as horizontal and vertical mixing very well (Swingedouw et al. 2022)." It would be nice to read if, how and why resolving eddies might make models more prone to tipping.**
We have expanded this description to clarify why this might lead more easily to tipping points of e.g. the SPG through its influence of Greenland meltwater spread.

**Fig. 4: Is there a reference for the experiment and the results? Is this an overshoot in $CO_2$ because emissions stop, or an overshoot because of negative emissions? Does the AMOC have a delayed recovery or alternative states in this model?**
The figure is adapted from Figure 17, Heinze et al., 2023 (in review). The emission scenario is SSP534, derived following the CMIP6 protocol for CDR-MIP (Carbon Dioxide removal Model Intercomparison Project, Section 4.6.2.1 in Lee et al., 2021). Here, delayed recovery is the main mechanism. As outlined in the main text, bistability cannot be concluded from the runs until 2100. However, two hysteresis loops are evident in the 2300 runs that might be indicative of bi-stability with the same AMOC strength but different T and $CO_2$ concentrations. These unpublished results are not part of this paper though (pers., comm, Rynders and Aksenov).

**I 209-210: "Although the AMOC does not collapse in this model, it seems unlikely that it will recover its former strength on human timescales." Why does this seem unlikely? What's the evidence? And what are "human timescales"?**
This has been rewritten.

**I 215-216: "while if it is preceded by…" what do you mean, a feedback being preceded by another?**
Yes this was meant, we have rewritten this part of the paragraph to make this more clear.

**l 240: "we may be close to an AMOC tipping point (Michel et al., 2022), as do the studies of Boers (2021) and Ditlevsen and Ditlevsen (2023) cited above". Not really; for example Boers (2021) only argues that the AMOC shows slowing down, but it does not make statements about whether a tipping point exists and how close it is. And Ditlevsen and Ditlevsen (2023) assume that a tipping point exists, and only determine its proximity based on this assumption.**

Thank you for reading carefully - we agree with this point of view and have corrected those sentences to be more cautious in our statement.

**l 248-251: "More paleo-reconstructions of AMOC strength, ocean surface temperature, and other AMOC-related properties with high temporal resolution, using appropriate proxies and careful chronological control performed for key past periods (e.g. last millennium, millennial-scale climate change events, previous interglacials), hold great potential to improve our understanding about the AMOC as a tipping point." I (and possibly readers) would be curious to learn more – why is there a great potential, what needs to happen, what are the proxies?**

We have expanded this part to provide a few more substance to those statements.

**l 252: "develop improved metrics" means AMOC fingerprints? This term is used in Sect 2.1.1, but then dropped.**

We have changed this to 'fingerprints'

**Fig. 6a: Caption says 2020-30, but figure says 2020-39. Both is in the future, but "projection" is not in the title, in contrast to the other three subfigures. Why? b) "changes ...by…" compared to what reference period?**

Thank you for spotting this, indeed panel a is not properly labelled; we have fixed the typo (39 instead of 30) now. Panel a serves as the modelled state for this decade, whereas the other panels show how projected changes by end of the first half of this century. As all the panels reflect modelled future states (unconstrained by data), we have now adopted a uniform labelling across all of them.

**l 526: potential tipping behaviour in the AMOC (relation to global monsoon described in West African monsoon below) or increase in the interhemispheric asymmetry of aerosol loading in the atmosphere beyond potential threshold levels could lead to large disruptions to monsoon systems.**

Not sure what the question or request is here, so we left this as is.

**l 569-570: "a process sometimes referred to as "induced tipping"." Any reference?**

We now added a reference here (Pausata et al., 2020)

**l 607: "in one model" – but not an Earth system model. Also, the concern about deforestation and moisture recycling in general, which coupled the SAM and the rainforest, somehow comes out of the blue sky here, and would deserve 1-2 extra paragraphs.**

We added more information to this paragraph.

**l 612: "see 1.3.2.1 for more on Amazon dieback" This section does not exist.**

Indeed. The text "(see 1.3.2.1 for more on the Amazon dieback)" was deleted.

**l 625 "low emissivity for longwave radiation (heat)," - do you mean transmissivity, or really emissivity? Low compared to what? Probably to thicker clouds. But compared to no clouds, the emissivity is high. And why do you compare high thin clouds with low thick clouds? Because these combinations are most common? With "high" vs "low" you mean their altitude, not their thickness? What about high thick clouds, like cumulonimbus?**

These were taken as two opposite examples to highlight the main features (altitude ~ temperature ~ longwave radiation emission; and thickness ~ shortwave radiation transmission). We have now rephrased to make this more clear (and corrected emissivity for emission).

**l 637: "the transition of shallow cloud layers from closed to open-cell geometries" How would the dynamics in droplet growth describe in the previous sentences lead to that? 1-2 sentences to explain the connection would be helpful.**

We have added some more explanation and an additional reference for more context.

**l 693-695: can now be updated with newest observations**

Updated with most recent observations up to September 2024.

**l 772-773: "Models with a strong AMOC reduction in the future tend to project a much stronger poleward shift of the jet than models with a weaker AMOC reduction" Why does this happen?**

This is because the AMOC reduction tends to enhance the latitudinal temperature gradient in the North Hemisphere, by warming the equator and cooling the northern high latitude. This increase in latitudinal temperature gradient tends to strengthen and shift the jet stream to the north. This is an effect sometimes referred to as the Bjerknes effect, by which the atmosphere somehow compensates for the decrease in meridional ocean heat transport, through jet stream enhancement and northward shift in the mid to high latitudes (e.g Bellomo et al. 2021, Swingedouw et al. 2009. We have added a short description of this.

**l 777: "Arctic is warming more rapidly than the rest of the planet, partly driven by sea ice loss" – sea ice loss (albedo feedback) is indeed not the only reason; maybe cite more comprehensive studies like Pithan & Mauritsen 2014**

We agree that sea-ice loss encompasses many more processes than only albedo - notably radical changes in energy and other transfers between the atmosphere and oceans, and our general reference to sea-ice loss was meant to go beyond purely the albedo feedback. We rephrased and added a reference to a recent review paper (Previdi et al., 2021).

**l 784-794: The explanation is not quite clear to me, how the jet stream could undergo tipping. The resonance behaviour seems to be the positive feedback that pushes the jet stream to another regime? Again, this is a paragraph where the paper would benefit from more explanations and interpretation, beyond describing results from the cited papers.**

We rephrased parts of this paragraph to highlight more clearly the following logical argument: warmer Arctic → more jet waviness → potentially weaker separation between high and low-latitude airmasses and further weakening of the Arctic. The last point is not

well-established in the literature, so we will present it as a hypothesis that we make by building on available evidence.

**l 801-804: Unclear what the connection between these sentences is. If the models are so uncertain, what evidence is it that some show tipping behaviour in blocking? And why "in addition"? These sentences make opposite statements in some way.**

The aim here was to express that the current literature offers little conclusive evidence for the existence of tipping points in the mid-latitude atmospheric circulation, but that we should not necessarily take this as evidence of lack of tipping. Indeed, the large uncertainties in how models represent tipping prevent us from equating a lack of evidence for tipping to evidence of a lack of tipping. We rephrased the text to express this notion more clearly.

**l 903: Why is the process complexity a "conceptual issue"? Conceptually, feedbacks are well defined, I'd rather say that the complexity is a practical limitation?**

Agreed that individual feedbacks are well defined but we do not label process complexity as a conceptual issue. There can be practical limits for conceptual efforts - in defining what is "relevant" for the system under scrutiny, and identifying the individual feedback loops and interaction with ambient systems. Similarly there can be practical limits for numerical exercises - when trading of resolution and type/number of represented processes against computational runtime. We have slightly modified the text.

**l 916 and elsewhere: biased (with one s, not two)**

Thank you for catching that, fixed in the resubmitted text.

**l 932: The paragraph on TIPMIP reads like an afterthought that is only there to mention the project. It could be either removed, or better integrated in the rest of the paper.**

The paragraphs right before that allude to shortcomings in present modelling and to dedicated ongoing and planned numerical experiments and community initiatives scrutinising the models' capacity to answer tipping points-related questions. This is a part of the scientific outlook given in the concluding section of the paper. We therefore considered the mention of TIPMIP as an international collaboration to streamline these activities quite fitting at this position in the paper. We have added some rephrasing now, hoping that this is more convincing now.

**Table 1: Why arrows up and down instead of + and - signs for positive and negative feedbacks?**

In this table, we reserved +++/--- for the uncertainty assessment, and used ↑ and ↓ to differentiate from that.

**Referee #2**

Thank you for investing the time and effort to review this manuscript - especially given its length and range of discussed systems - and for your thorough and critical assessment. Below, we address each of your comments individually.

The paper provides a well-written and comprehensive review of tipping points in global climate subsystems including ocean circulation and the monsoon systems. For each of the discussed subsystems the authors provide a paragraph on the evidence and a paragraph where the evidence is assessed with respect to potential tipping behavior. I find the assessment generally well-balanced. A complication in the discussion on tipping points is that there are very few observations of tipping from the instrumental record and hence paleoclimatic evidence must be used. As outlined below, I feel that the authors should specify why they see proxy records related to ocean circulation as evidence for tipping dynamics in the AMOC. For me this is far from obvious. The paper would also benefit from a better description of the feedbacks involved in modulating the West African Monsoon, particularly the role of the tropospheric circulation.

1) Definition of tipping. What is the definition of "tipping" employed here? Some of the classical definitions of abrupt climate change (response faster than the forcing, rate of change only determined by the climate system, not by the forcing, difficulty of eco-/economic systems to adapt, etc. see for example National Research Council, 2002) are difficult to apply to proxy records. Please be more specific what qualifies the AMOC as a tipping element based on paleoceanographic data.

We have now added our working definition of tipping in the introduction, as well as a more transparent overview on the criteria we applied for the classification as a tipping system.

2) AMOC as a tipping element. There is no question that there is evidence for significant AMOC variations with huge consequences for climate and ecosystems. However, if the AMOC is actually a tipping element is much harder to determine and I would argue that there is not much unequivocal evidence for tipping dynamics in the AMOC. For example, Pa/Th, $\partial^{13}C$ and flow-speed related proxies do not show any indication for an abrupt decline in AMOC into HS1 (e.g., Stanford et al. 2011). Apparently, the slowdown is quite gradual and takes several millennia, inconsistent with the time scale of tipping of 15-300 years mentioned in line 212. By contrast, the onset of the AMOC (e.g., with the Boelling/Allerod interstadial or DO1) seems often indeed very abrupt and much faster than the slowdown. One could argue that the AMOC resumes almost instantaneously with the end of the anomalous meltwater flux (see for example $\partial^{18}O$ in Fig 4f in Stanford et al. 2011) at the end of the Heinrich Stadials, which might hint against a strong bi-stability of the AMOC and for a more linear relation between AMOC strength and freshwater flux as suggested by Liu et al. (2009). So again, in the light of conflicting results from climate models, how can we derive from proxy records that the AMOC is actually a tipping element? I am not ruling out that the AMOC is a tipping element, but is there enough evidence apart from some (mostly intermediate complexity) models to qualify the AMOC as one with 'medium confidence'? Since this is an important question, we should be careful with the answer. Even a small and more gradual AMOC change might be very dangerous in certain regions of the world (see comment 4).

Recently hysteresis simulations have been performed with the CMIP5 version of the CESM and with the strongly-eddying version of the POP ocean-only model. The CESM definitely has a multi-stable AMOC regime (Van Westen & Dijkstra, 2023) and the AMOC collapses in the POP model. Several CMIP6 models have also shown the presence of a weak AMOC

state that can persist for over 100 years. Hence, AMOC tipping is now found over almost the full hierarchy of models. This is added to the revised text.

On the paleo-climatic evidence, we agree that AMOC slowdown at the onset of Heinrich stadials apparently takes longer than 15-300 years. However, regarding the AMOC recovery at the end of Heinrich stadials, the mechanism proposed by Liu et al. (2009) (i.e., end of the anomalous freshwater flux to the high latitudes of the North Atlantic) is not the only possible one. At least two other mechanisms have been proposed that are not incompatible with a hysteresis behaviour of the AMOC: (i) a decrease in salinity of Antarctic Intermediate Water due to a predominant southern source for meltwater pulse 1A (Deschamps et al., 2012); and (ii) an increase in the inflow of high salinity surface and central Indian Ocean waters into the South Atlantic via the Agulhas Leakage (Chiessi et al., 2008). Thus, the AMOC recovery at the end of Heinrich stadials is not negating a tipping dynamics for it.

Overall, we nuanced the phrasing taking the points above into account. Based on the paleo-proxy and model evidence (across the hierarchy), we stand by our classification of AMOC as a potential tipping system with medium confidence, and hope that the manuscript updates reflect this accordingly.

**3) Collapse of AMOC (Line 126 "It also occasionally collapsed to an off-mode"): The recent literature does not support the existence of an "off-mode". The recent assessment by Pöppelmeyer et al. (2023) finds a reduction of the AMOC of about 30% relative to the LGM during HS1. There is also proxy evidence for NADW formation during HS1. For example, based on εNd, Howe et al. (2018) find no evidence for strong changes in water mass provenance in the mid-depth South Atlantic between the LGM and HS1 and benthic isotopes support active deep-water formation in the North Atlantic during HS1 (Repschläger et al. 2021).**

We agree that this point deserves a more detailed description, but disagree that the existence of the off mode is not at all supported. Many authors have concluded that 3 modes for the AMOC exist and that the HS1 consists of 2 stages H1.1 and H1.2 with probably a two-step change from warm on to cold on to off. We agree that switches between warm on and cold on appear to have occurred more frequently in the past, but also do not see why this switch would not be a tipping point. See also the discussion in Weijer et al. 2019. We have adjusted the phrasing here.

**4) AMOC and WAM during the instrumental period. Personally, I have no confidence that the AMOC is actually a tipping element because both the palaeoceanographic data and the models are ambiguous. The focus on "tipping" might distract from the fact that even comparably moderate fluctuations in the AMOC can have significant and dangerous effects on ecosystems, economy and society. In this context, the paper would benefit from expanding to the very few examples of abrupt climate change in the instrumental record. The event that that is often cited as the most recent example of abrupt climate change is the abrupt onset of the multi-decadal catastrophic Sahel drought in the early 1970ies (briefly mentioned in line 546) with an abrupt reduction of precipitation on the order of 30% over nearly two decades and probably millions of victims. Lake Chad shrank by >90% over the following decades. Initially, it was thought that overgrazing and desertification was to blame for the drought (Charney, 1975), but subsequent work made it clear that the drought is connected to a specific SST pattern with a negative SST anomaly in the North Atlantic and positive SST anomaly in the South Atlantic (Folland, et al. 1986, Bisautti, 2019,**

Pereira et al. 2022), an SST pattern we know as bi-polar seesaw from ice cores, sediment cores and climate models as a response to the weakening of the overturning. It seems therefore likely that the 70ies Sahel drought is connected to a fluctuation of the thermohaline circulation as various authors suggest (e.g., Knight et al. 2005, Zhang and Delworth, 2006) and that the Charney-Effect is only second-order feedback. If the WAM is indeed a tipping element, why did the precipitation and the vegetation of the Sahel immediately recover (e.g., Heumann et al. 2007) after the drought with the onset of the positive phase of the AMO, although precipitation and vegetation were significantly reduced over decades? The Sahel likely turned into a complete desert during HS1 (there is evidence from fossil "Ogolian" dune fields, Collins et al. 2013), but it recovered synchronously with the onset of the Boelling interstadial, again not much evidence for bifurcation or an irreversibility in the African Monsoon, both on the millennial scale and the decadal scale.

Thank you for this perspective, we agree about the reversibility but would like to point out that abruptness and irreversibility, while often consequences of tipping, are not inherent to our tipping definition (consistent with the prevailing approach in Earth System science, see references in introduction). Rather, we anchor our analysis on the identification of self-sustained positive feedbacks. In this case, the albedo effect introduced by Charney plays a key role that amplifies and sustains the drought following a critical threshold (which might be transgressed/initiated by changes in SST).

I agree with the authors that AMOC reconstructions are highly uncertain. But it is evident from both data and models that the ITCZ/tropical rainbelt is very sensitive to changes in AMOC intensity on all time scales (Marshall et al. 2014, McGee et al. 2014). If the AMOC is already slowing down, this raises the question why Sahel precipitation and vegetation has increased in recent decades. In my view this shows that the AMOC was indeed dominated by multi-decadal variability as suggested by Latif et al. (2022). Based on the work of Wett et al. (2023) it is probably robust to say that there are no trends in the instrumental record of AMOC observations since 1993.

We agree that there is a lot of uncertainty in AMOC reconstructions, and that the attribution of significant trends in instrumental records is contentious (possibly because the large variability is obscuring a weakening), see discussion in main text. With respect to Sahel precipitation, it is possible that other changes in climate are also affecting this region, and a detailed investigation is out of scope for this study.

5) Monsoon, ITCZ and AMOC. The response of the Monsoon to AMOC slowdown is portrayed as a simple southward migration of the ITCZ. This is certainly a good model when it comes to areas primarily influenced by the clearly defined oceanic ITCZ, for example NE Brazil. However, it seems to me that this model is an oversimplification when it comes the monsoonal areas in Africa. Some meteorologists even argue that the ITCZ and the rainbelt located between the African Easterly Jet and the Tropical Easterly Jet (where 50 - 80% of the rainfall is produced by a relatively small number of mesoscale convective systems) are different systems and should not be confused (Nicholson, 2009, Fig. 18). If the response would be a simple southward migration of the rainbelt/ITCZ over Africa, we should see regions to the south of the ITCZ, where precipitation increases. For Africa and the Monsoonal areas this seems not to be the case (e.g., Stager et al. 2011). More recent work points to the location and intensity of the tropospheric jet streams as important processes modulating the strength and

**position of the African rainbelt (e.g., Farnsworth et al., 2011, Nicholson and Dezfuli, 2013). This should be mentioned in the paper.**

Thank you, we have included mention of this more recent work in the revised manuscript in the discussion of the impact of AMOC and other external drivers on potential monsoon shifts. Please note that in the cited reference Stager et al. (2011), the weakening of the AMOC is associated with a cooling of tropical oceans, which in turns seems to be responsible for a general weakening of the global monsoon. The importance of SSTs, in particular in the Indian ocean for the ISM, is demonstrated in Pausata et al., (2011).

**6) Missing info on AMOC variability during interglacials. Much of the cited evidence for AMOC variability comes from the glacial period. In this context it might also be important to mention the 8.2 kyr Event as an example for an AMOC slowdown under interglacial boundary conditions. In Africa, this event was associated with aridification as documented by low lake levels (Gasse, 2000) and periods of human abandonment in the southern Sahara (Sereno et al. 2008).**

We have drawn on Galaasen et al. (2014) to show signs of instabilities for the last interglacial (Fig 3), and now explicitly mention the 8.2kyr event in the revised manuscript.

**Stefan Rahmstorf**

Thank you for investing the time and effort to review the AMOC part of our manuscript. Below, we address each of your comments individually.

**This is generally an informative review worth publishing, but I'd like to flag a couple of issues which should be improved, or else may lead to misunderstandings.** *"AMOC bistability is model-dependent though, controlled by the balance of the positive and negative feedbacks that determine the salinity of the subpolar North Atlantic. It is not yet understood why the bistability occurs in some models and not others (Jackson et al., 2023)."* **This statement seems to mix up models *having* a bistable AMOC regime, and models *being in* this bistable regime for present climate. As explained earlier in the article, it requires a special hysteresis experiment to test whether a bistable regime exists, and as far as I am aware every single model which has been tested in this way does have a bistable regime (the latest example being van Westen et al. 2024). So, as far as we know this bistability is not model-dependent but a very robust feature across a wide range of models from Stommel's simple box model to modern climate GCMs. What Jackson et al. 2023 have shown is merely that some models are not in this regime for present climate, a finding consistent with many other previous studies. That is not a fundamental model difference but a matter of tuning and accurate representation of salinity. However, the wording quoted above wrongly suggests that some models don't have a bistable regime. This must be clarified.**

Agreed and the wording is changed to better reflect this. There is now also a clear example from a low-resolution ocean-only model that  a model bias (in this case a freshwater bias in the Indian Ocean) shifts the AMOC tipping points (Dijkstra and Van Westen, 2024). In this case, indeed the present-day state may shift into the single equilibrium regime, but this does not mean that the model does not have a multi-stable regime.

"It is therefore difficult to confidently discern potential recent trends from natural variability, due to disagreement between published studies (Bonnet et al. 2021, Latif et al., 2022, versus Qasmi, 2022)." I do not see disagreement between these studies regarding recent trends; rather this again mixes up different issues. Qasmi 2022 indeed analyzes recent trends in observational data (with the help of model simulations), namely the Atlantic 'warming hole' - and comes to the clear conclusion that it is anthropogenic. The same conclusion was reached earlier by Chemke et al. 2020 (which should be cited): "Analyzing state-of-the-art climate models and observations, we show that the recent North Atlantic warming hole is of anthropogenic origin". Latif et al. merely analyse CMIP6 models. As the IPCC has shown (figure SPM.5a of AR6 WG1 report), these models overall do not reproduce the 'warming hole' until the present and don't show an AMOC weakening until now, only in future. An important finding of Latif et al. is, however, their Fig. 4 showing how the actual AMOC in CMIP6 models is correlated with the AMOC SST fingerprint of Caesar et al. 2018 (both cold and warm part), which supports the conclusion that the anthropogenic 'warming hole' discussed by Qasmi and Chemke indeed points to an AMOC weakening. (Bonnet et al. 2021 is also not in disagreement with either of the cited other two studies but looking at a different aspect again.) So there is no disagreement between these studies, but rather a model-observations disagreement which is very important for the tipping point risk discussion: the data suggest an anthropogenic AMOC weakening already in recent decades, which the CMIP6 models do not reproduce. Which suggests that the models understate the slowdown (and thus tipping risk). That echoes my first point, where many models are not in the bistable regime but observational data suggest the real AMOC is in the bistable regime, so the models are likely too far way from the tipping point.

We agree that this statement was not clear enough, since this is a very slippery topic that can easily lead to misunderstanding. We rewrote it in order to say that, on the one hand, a weakening of the AMOC over the last century can be explained by internal variability in a number of models (Bonnet et al. 2021, Latif et al., 2022), and that, on the other hand, if such a trend has been forced by external, then models do not capture it correctly (Menary et al. 2020. Concerning the SST trend in the subpolar gyre, if Chemke do show that this region is experiencing less warming than elsewhere in climate models (so that we can name it warming hole), the forced response of this region to all external forcing in CMIP6 models is that of a warming (cf. Figure 2 of Qasmi et al. 2023). Thus, from this evidence, we cannot properly attribute the cooling trend in the subpolar gyre to anthropogenic forcing. To avoid adding confusion by citing the paper from Chemke et al. (2020) that focus on the relative warming of the North Atlantic compared to the rest of the world, and not on the AMOC (a word not used in their paper), we prefer to avoid citing it.

To sum up, readers must not be confused with messages like "some models show bistability, some don't" and "some studies suggest recent anthropogenic AMOC weakening, some don't" which are not backed up by a careful reading of the cited evidence. One more point: "However, the proxy data used in these studies have large uncertainties, and some other reconstructions show little evidence of decline (Moffa-Sanchez et al., 2019, Killbourne et al., 2022)." When you cite the comment by Kilbourne et al., please also note our reply to their comment, particularly our Fig. 2 which shows there is a high consistency amongst the reconstructions: https://www.nature.com/articles/s41561-022-00897-3

Thank you, we have amended this section accordingly.

---

## Author Response (AR2)

**Author's response to "Tipping points in ocean and atmosphere circulations"**

We thank Axel Kleidon for handling our manuscript and the reviewers for their feedback to improve our manuscript, which we have revised accordingly. Below are the original comments from reviewers in black and our corresponding responses in blue.

Sina Loriani for the author team

**Reviewer 1**
To recap, my main suggestions in the previous review stage were:
• 1. go deeper into mechanisms of circulation "tipping" and thus go beyond previous reviews; include East Asian monsoon
• 2. clearer and more objective method and criteria of selection of cases, and labeling of uncertainties
In general, the authors have implemented some changes in the direction of these suggestions. I however also see the need to go a bit further, and have some open questions that I will address below. I do not think that the article has to become substantially longer, but it should be rephrased accordingly; and/or some parts (especially impacts, which are not part of the confidence assessment) may also be shortened.

We'd like to thank you for reviewing this – rather extensive – article again, and for the constructive feedback! Especially the reflections regarding more transparency on definitions and criteria were helpful to make sure that the article provides a solid reference for a wide readership. Following your suggestions, we have amended the introduction to provide a clearer definition of "tipping", and to describe the methodology that the confidence statements are based on more transparently. Furthermore, we have added a dedicated section on East Asian monsoon. Point-by-point replies to your comments can be found below.

I also urge the authors to make more transparent what has changed between versions, and what not. I also notice that the "original" (black text) in the difference file does not agree with the previous version I reviewed. This makes it very hard to see what has changed. Often, the information in the replies was misleading (changes the authors claim to have made are not visible in the pdf I got, and pieces of text were hard to find because neither line numbers or quotes were provided).

Thank you for pointing this out! Indeed there seems to have been an unfortunate mixup on our side (some changes that indeed were not colored/implemented in the version of the submitted manuscript), apologies for that. And, at the same time, apparently there was an issue on the ESD-server side (?), since some of the changes you remarked as not being implemented *are* in fact addressed in the track-changes-file we uploaded [egusphere-2023-2589-ATC1.pdf](). In any case, apologies for the arisen inconveniences – we double checked that the now resubmitted file highlights all changes. All the highlighted differences are with respect to the *first* submitted manuscript.

General
1. I still suggest to discuss the EASM in its own (sub)section (see below).

We have now added a dedicated section on East Asian monsoon.

2. The confidence levels are now more explicitly explained, but still a bit vague. Are conceptual hypotheses "evidence", and how do the authors weigh results from conceptual models versus ESMs? How many models, and models of what sort, are required to show "tipping", for which forcing, in order to go from a "+" to "++"? I understand that this will always remain subjective to some extent, but I believe that some more precision is possible here. The resulting confidence assessment should be the clear result of pre-defined criteria; otherwise the criteria can appear to be "back engineered" to the subjective opinion of the authors of the (not peer reviewed) "Tipping Point Report".

We have added a dedicated subsection in the introduction providing more context to our criteria. In short, we anchor our assessment on the question "*is there a plausible positive feedback to drive self-perpetuating change beyond a forcing threshold*" and review the amount and agreement of different lines of evidence to support or refute the existence of such a feedback. This links to the discussion of the tipping point definition (see your comment and our response below).

We here approach each system with the hypothesis of such thresholded feedback dynamics (often guided by physical process understanding and conceptual or simplified models). Our confidence to classify a system as a tipping system increases with the amount and agreement of high-quality evidence from the different sources (models, paleo records, observations). Due to the complexity of the problem, at least with present-day literature, it is very challenging to make a generalised link between a confidence statement and a number of models to exhibit tipping dynamics. First, one needs to know whether these models do resolve the key feedbacks (i.e. *can* the suspected tipping process be studied with the models in the first place?) and second, generalised statements need a sufficient number of such models to participate in a model intercomparison designed to answer tipping-related questions. The emerging generation of CMIP7-type models, as well as initiatives like the Tipping Points Modelling Intercomparison Project (TIPMIP) and related endeavours are expected to provide more solid grounds for future assessments that could attempt to relate confidence statements to some quantified metric, like fraction of models exhibiting tipping dynamics in comparable settings. In the meantime, we rely on expert judgement classifying the consistency of evidence across different types of models and different experimental settings.

3. The authors replied to my previous points and often elaborate on some research results in their replies, but often left the actual manuscript unchanged. The readers of the article may benefit more if some of these replies make it into the paper. For example, can we expect some systems to be more prone to tipping than others? Are clouds not too fast and variable to be an Earth system tipping component? How does resolution affect tipping behaviour? What needs to happen to reduce uncertainties?
For example, the authors replied: "We agree that more theoretical insights from hydrodynamic theory such as critical transitions in turbulence on multiple scales (van Kan and Alexakis, 2020; van Kan, 2024) or transitions between multiple- stable states in aquaplanets (Brunetti et al., 2019; Ragon et al., 2022) can inform a better understanding of more applied research on climate tipping points going forward. We will add a corresponding discussion and literature references to the revised manuscript." However, I do not find such a

discussion in the revised paper, and do not find citation of the papers they mention. Brunetti et al. and Ragon et al. both appear in the reference list without being cited in the text, the others do not appear.

More paragraphs that the authors claim to have revised, show no changes. Examples of unchanged(?) text:
• My previous review: "Also apparently unchanged: Sentence is unclear. What does "these" refer to?" Reply: "Meant were small-scale processes, added." I see no change.
• "Although the AMOC does not collapse in this model, it seems unlikely that it will recover its former strength on human timescales." Why does this seem unlikely? What's the evidence? And what are "human timescales"? Authors: "This has been rewritten." I see no change.
• I suggest the authors go through my previous review and highlight the actual changes made.

We now transferred some of the responses to the main document where it seemed fit well without expanding the text too much; mainly in the discussion section (marked in blue). On the lack of change in the previously revised manuscript, please see our comment above.

Abstract
• nothing seems to have changed? I suggest to make clearer what the approach is in this paper, in the direction of my main suggestions above.
• specifically: "we classify the West African monsoon as a tipping system." sounds like you don't consider any other system. Clarifying the method and criteria of the classification would help, and make clear that highlighting the West African monsoon is the result of this method?
• "modified wind patterns...disrupt established circulation patterns" sounds redundant - what are drivers of such shifts?
• The authors write in their reply: "The sentence aims to highlight the complex interplay of factors affecting monsoon stability. While the AMOC is not a direct driver of monsoons, research suggests its variability can influence large-scale atmospheric circulation patterns that, in turn, impact monsoon systems. We have now rewritten the sentence to reflect this more clearly." In the files I received, nothing has been changed in the abstract. The sentence is still there. In general, it would help if the authors quoted from the new version in their replies, referencing line numbers, to substantiate claims about what they have changed.

We have reworded the abstract to make clear that the reported classifications are based on the evidence found in the available literature and resolved the ambiguities you indicated above.

Introduction
• The authors added more text on the definition of "tipping"; but some of my detailed comments remain, e.g. "tipping" in general, or due to human activity in the future? Why focus on circulations, what are open questions there? How was literature collected? More precise assessment criteria.

Thank you, this is indeed important to clarify. We are analysing systems that have been fairly stable under preindustrial/Holocene conditions, and where it is plausible that they change under human activity. The guiding question is therefore: *Which components of the Earth system, with focus on circulation systems, feature strong feedbacks that could lead to*

*self-sustained change after crossing a forcing threshold*? In different words, "which systems *could* in principle tip?" We'd like to highlight (in agreement with your comment below) that this is a different question from "which systems *will* tip", which is subject to current/future forcing trajectories (and uncertainties thereof) and the location of the forcing thresholds (and uncertainties thereof). The confidence statements in this review refer to the former, i.e. *could* the systems tip.

This work focuses on circulation systems as part of a special issue. It is therefore complementary to analogous assessments for systems in the cryosphere and biosphere, and provides a detailed deep-dive into the selected systems that goes beyond recent reviews addressing systems from all domains (Wang et al., 2023; Armstrong McKay et al., 2022). Circulations in the ocean and atmosphere critically connect different parts of the Earth system and hence play a vital role e.g. in potential tipping cascades (Wunderling et al., 2023).

For this narrative review, literature was collected via expert elicitation among the group of authors. We describe the confidence assessment framework above, and in the newly added dedicated subsection in the introduction.

• Definition of tipping. The authors adopt a definition that involves positive feedbacks, and even "self sustained" feedbacks. I believe that is OK as it is made transparent, but please resolve these potential issues:
• 1. There are other definitions around. Specifically, it would make sense to also note that the IPCC actually gives a different (less specific) definition; see AR6 glossary. The part "often abruptly and/or irreversibly" is actually a quote from IPCC that should be referenced. It seems that the authors decide to use a narrower definition than IPCC because positive feedbacks are essential here; this should be mentioned.
• 2. Do all examples really adhere to your definition? Coral bleeching: Why does it crucially involve self sustained positive feedbacks? As far as I know, corals die when temperature becomes too high, without the need of runaway feedbacks? Same goes for permafrost (the global permafrost - CO2 feedback is far too weak for self-sustained feedbacks).
• 3. What exactly do you mean with "self-sustained feedbacks"? If a forcing is faster than the system's response (a typical situation in climate change), would not any system change involve "self-sustained feedbacks" as long as the system tracks the changing equilibrium, even if the system is perfectly linear?! Fig. 4 could be an example of such a situation. The problem of such a definition based on feedbacks is also that you need to understand well why a transition occurs in a model, rather than just observing that it does. Uncertainty about mechanisms hence leads to lower confidence, even though the phenomenon may be apparent.

The revised introduction has a subsection now on the tipping points definition we are adhering to, which is based on f*eedbacks strong enough to sustain change after crossing a forcing threshold* (i.e. irrespective of continued forcing).

We argue that our definition is compatible with the one from IPCC, where a tipping point is defined as *the critical threshold beyond which a system reorganizes, often abruptly and/or irreversibly* (4.7.2 in Lee et al., 2021 / IPCC AR6 WG 1 report Chapter 4). However in the IPCC, criteria for the system's reorganisation are not defined, rather irreversibility and

abruptness are used as proxies for tipping, leading to sometimes inconsistent results such as ocean heat content being listed in the tipping elements table 4.10 of that chapter despite a threshold-free behaviour. These and similar arguments (e.g. on the ambiguity of defining *abruptness* in this context) have been made in Armstrong McKay et al. (2022), and we have included a corresponding remark in the introduction.

By extending the IPCC definition with the criterion of *feedbacks sustaining change beyond a threshold*, we require some understanding of why/how the system reorganises after crossing the (tipping-point-defining) critical threshold. Arguably, rather than constituting a problem, such a definition provides a generalisable yet consistent approach to tipping points, and can work across a wide range of systems as recently demonstrated in the Global Tipping Points Report (2023) co-authored by more than 200 experts in the various domains (including assessments of coral reefs and permafrost with the same approach – in both cases *local* rather than global feedbacks are assumed to drive tipping). On your last point; we hope that the extended tipping point definition in the introduction (and references therein) explain better now what we mean with feedbacks that sustain change after a critical forcing threshold.

• 4. It makes a difference in confidence whether a system can in principle show tipping behaviour, e.g. has shown such behaviour in the past, versus whether this system will show "tipping" as response to plausible human forcing in the future. The kind of evidence you cite and the confidence levels seem to imply the former ("system features tipping dynamics", e.g. paleo transitions), but the context of the article and the definition you adopt from McKay et al., Lenton et al., ... follows the latter. For example, if there is evidence for AMOC tipping during ice ages, what does a "++" assessment mean for the possibility of human-induced AMOC tipping? This should somehow affect the assessment, or the interpretation of the assessment.
• The different confidence levels all seem to repeat the same phrase; no need for such repetition!

As described above, we here focus on which systems *could* in principle tip (more formal: Is there a feedback that could lead to self-sustained change after a forcing threshold) and not on whether these systems *will* tip (more formal: Will this threshold be crossed). We would kindly disagree that adopting the definition from Armstrong McKay et al (2022) and related literature implies that we consider the latter. In the context of our article, "++" means that there is medium confidence in the potential for the AMOC to tip, with uncertainties in potential timing (related to threshold position), magnitude or feedback strength.

Sect. 2

• line 182: "Here we assume that all three states (Alley et al., 1999; Rahmstorf et al. 2002) are stable equilibria" Why? Is it necessary to make such an assumption in a review paper?

Agreed that this wording was misleading, it was rather a comment on the underlying conceptual three-mode model. We have rephrased this.

• SPG: Why medium confidence, the same as the AMOC? I get the impression that some ESMs do show AMOC "collapse" in some way, but are there such ESM results suggesting a

separate collapse of the SPG (independent of AMOC)? Is Fig. 6 showing "mixed-layer depth" supposed to show such an SPG collapse? Does the gyre (horizontal circulation) actually collapse?

• Same question about the Southern hemisphere. There seems to be less model evidence and agreement; yet all three ocean systems get a "++" in the confidence assessment. I guess the issue here lies in the criteria that are still a bit vague, see above.

On the SPG: This system is linked but independent from AMOC. E.g., Sgubin et al. (2017) show that there are CMIP5 models where the SPG convection collapses both with and without an accompanying AMOC collapse. We made small amendments to make clear that the SPG is driven by both wind shear and buoyancy forcing – the latter being subject to a potential *SPG convection collapse* that would reinforce (and vice versa) an overall *SPG weakening* (horizontal circulation slowdown), see also Sgubin et al. (2017) and Swingedouw et al. (2021). Mixed layer depth is an indicator for the convection. We added a corresponding statement.

On the Southern Ocean circulation: As described above, approaching each system with a conceptual understanding of a potential tipping mechanism, model evidence needs to be scrutinised with respect to the capacity of the model to resolve this mechanism in the first place. While this process understanding has long been limited for the Antarctic Overturning Circulation (Purich and England, 2023), recent work demonstrated that under adequate modelling conditions (e.g. representation of deep water formation over the continental shelf), models do show a collapse akin to potential AMOC tipping (Li et al., 2023, Zhou et al., 2023), supported by paleo evidence (Huang et al., 2020 and other references cited in the manuscript) thereby adhering to the ++ criterion in our assessment.

On a more fine-grained scale, these three systems could arguably be separated more subtly but here for all three of them we argue that there is ample evidence to classify them beyond + and too little to classify them as +++. We hope that our refined assessment criteria presentation in the introduction convincingly supports this.

Sect. 3

I do not understand the argument why the East Asian summer monsoon should not be addressed in its own subchapter. The authors now mention this system, but only as an afterthought in Sect. 3.4 (Summary), although it did not appear in the sections that are supposed to be summarised here. The authors replied that the EASM is not a "tropical" monsoon, but they already also consider ENSO, and mid-latitude dynamics, and the ocean circulation in their paper. Given that the focus of the review is a "wide range of circulation systems" in general, and that paleo evidence suggests abrupt shifts in the East Asian summer monsoon, I don't see why this particular circulation system should not have its own (at least small) section.

We have added a dedicated section on East Asian monsoon.

---

## Author Response (AR3)

**Author's response to "Tipping points in ocean and atmosphere circulations"**

We thank Axel Kleidon for handling our manuscript and the reviewers for their feedback to improve our manuscript, which we have revised accordingly. Below are the original comments from the reviewer in black and our corresponding responses in blue. The line numbers refer to the resubmitted manuscript.

Sina Loriani for the author team

**Reviewer 1**
Overall, the authors have now thoroughly addressed my comments and have substantially revised their manuscript accordingly. I congratulate them for their achievement to write such a comprehensive review.

Thank you for the thorough and constructive feedback! We are quite grateful for the scrutiny (especially given the length of the paper) and appreciate how it helped to significantly improve the comprehensiveness of the manuscript.

I only have two comments that can probably be addressed easily:

1. Tipping as a system property versus scenario driven:
I believe the choice of the authors to consider the existence of any tipping point in principle, whether it is plausible to be reached by future forcing or not, makes sense. Thank you for clarifying. Despite this convincing reply, I did not see this mentioned in the new manuscript, which would mean that this clarification is lost to the readers. I suggest to add it (unless I overlooked something).

Thanks for catching that – we added a short note in the introduction (line 132).

2. Definition of tipping:
a) The authors now refer to "the IPCC definition" (page 4), but it would help to state what it is. Their definition given above includes positive feedbacks, which are not part of the IPCC definition.

Agreed, we have added that now (line 69ff and 128ff).

b) The definition is now clearer. The authors appear to understand "tipping" as a phenomenon where positive feedbacks take over regardless of the trajectory / speed of the future forcing ("positive feedbacks change a system in a self-sustained fashion", "at a rate largely determined by the system itself").
This seems to match exactly the definition of a catastrophic bifurcation where a stable attractor disappears, a mathematically well-defined phenomenon. Then, the authors should call it by its name. However, they only mention bifurcations twice on page 11, without explanation of the term.

If the authors however wish to include stable transitions, with only one stable attractor that may nonlinearly shift with the forcing (my impression is that this is intended to be included in the McKay definition?): The dominating mechanisms that determine the system's dynamics

will then always be negative feedbacks, at least if the change in forcing is slow. In other words, in this situation, the notion that positive feedbacks determine the dynamics, does not appear adequate.

Thank you – we have adapted the text in the introduction where we define tipping points (line 69ff and 129ff).
* * *
Additionally to these reviewer comments, we made the following minor amendments to the last submitted manuscript:
- Updated affiliations and acknowledgements
- Added references to answer the reviewer's questions
- Added one extra reference for TIPMIP
- Minor rewording in the ENSO section (l 832ff)